# Has it *really* stopped? Interplay between rheology, topography and mesh resolution in numerical modelling of snow avalanches

Saoirse Robin Goodwin[1*,2*,3], Thierry Faug[1], and Guillaume Chambon[1]

[1]Univ. Grenoble Alpes, CNRS, INRAE, IRD, G-INP, IGE, 38000 Grenoble, France.
[2]Karlsruhe Institute of Technology, Karlsruhe, Germany
[3]Ove Arup & Partners Hong Kong Ltd, Festival Walk, Kowloon Tong, Hong Kong
[*]Formerly affiliated

**Correspondence:** Saoirse Robin Goodwin (srgoodwin@protonmail.com)

**Abstract.** Depth-averaged models of snow avalanches have hitherto lacked an objective arrest criterion. In this study, we investigate the stoppage mechanisms of simulated avalanches, considering the interplay between mesh resolution, simple/complex topographies, and cohesion. We use a second-order depth-averaged model including a modified Voellmy model with cohesion and a physical yielding criterion. Simulated results were found to be sensitive to the mesh resolution, until the cell width is less than 20% of the characteristic flow depth. The yielding criterion is sufficient to unambiguously define flow arrest for highly cohesive avalanches, even on the complex topography with feature sizes comparable to the typical flow depth. In contrast, for weakly-cohesive avalanches on the complex topography, a fully static state is never reached, due to numerical diffusion enhanced by local non-zero slopes. We hence investigated different global and local arrest criteria applied during post-processing, complementing the yielding criterion. Most of these criteria require setting ad-hoc thresholds, the values of which depend on numerous factors. However, tracking the evolution to a static state of the highest point of the flow material in the runout zone appears to offer an objective and practical solution to indicate when the model enters a numerical-diffusion-dominated regime, whereupon simulations can reasonably be terminated.

## 1 Introduction

Depth-averaged models closed by the empirical Voellmy friction law (Voellmy, 1955) can be solved through the Finite Volume Method (FVM) to model snow avalanches. Examples include RAMMS (e.g. Christen et al., 2010b; Frank et al., 2017; Martini et al., 2023), RASH-3D (Vagnon et al., 2019) SAMOS (Sampl and Zwinger, 2004), r.avaflow (Mergili et al., 2017), faSavageHutterFOAM 1.0 (Rauter et al., 2018) and TRENT2D (Zugliani and Rosatti, 2021), as well as the model used in the present study (Naaim et al., 2004). Numerical diffusion affects these numerical models, the extent depending typically on the numerical scheme adopted, the friction law used, and the topographical complexity (see also Hergarten, 2024). In practice, this numerical diffusion can lead to excessive spreading of avalanche deposits, smoothing of fronts, and, relatedly, persistent non-convergence of velocities to zero after apparent macroscopic flow stoppage. These artifacts are generally caused by numerical discretization of the flow domain and numerical handling of floating point numbers.

Natural Hazards and Earth System Sciences
EGU
Discussions

Velocity non-convergence consistently affects simulations involving complex topographies, even for high-order numerical schemes and high spatial resolutions. Correspondingly, automatic termination of simulations is generally based on subjective, ad-hoc thresholds. RAMMS2D, for instance, implements several options for user-defined thresholds to terminate simulations, the most commonly used considering the ratio between current global flow momentum and the maximum global momentum reached during the flow. Zugliani and Rosatti (2021) suggest using typical values between 1 and 10% for this threshold. Eckert et al. (2010) define a termination criterion considering the flow volume that continues to discharge, relative to the initial volume, adopting a threshold value of 0.01 %. However, such arbitrary thresholds may obfuscate analyses, e.g. when avalanches *almost* stop on an intermediate plateau, reaccelerating on steeper inclines downstream. The latter motion may not be reliably captured if overly high arbitrary thresholds are used.

For friction-dominated flows, Zugliani and Rosatti (2021) identify local transitions to *static* states using an *objective* stress-based criterion linked with Coulomb friction. Sanz-Ramos et al. (2023) implicitly allows flow material to become *static*, balancing resisting and destabilising forces when the velocity reaches zero, although this still requires defining "zero" velocity. These are essentially *yielding* criteria. Although ideally researchers and practitioners could adopt objective yielding criteria for defining simulation termination, waiting for all material entering a *static* state may not be computationally tractable. Consequently, *complementary* methods to define flow *arrest* are still required. Furthermore, the interplay between the *yielding* criteria and numerical diffusion was not specifically studied in Zugliani and Rosatti (2021) and Sanz-Ramos et al. (2023), so the robustness of these schemes with regards to, e.g., topography complexity and resolution, remains unclear.

Results from previous studies (Christen et al., 2010a; Bühler et al., 2011; Miller et al., 2022) demonstrate the interplay between numerical diffusion and spatial resolution for depth-averaged simulations on complex topographies. However, studies bridging simple and complex topographies are required to investigate this interplay further. Generally, the initial validation of numerical models is based on simplified topographies (e.g., inclined planes), with minimal numerical diffusion. Such validations are assumed to automatically qualify the models for applications to complex topographies (e.g. Christen et al., 2010b; Rauter et al., 2018; Mergili et al., 2017; Zugliani and Rosatti, 2021), without systematic sensitivity studies on interplay between e.g. terrain complexity and the performance of yielding criteria.

Although spatial resolution and topographical complexity both affect the ability of numerical models to handle stoppage, factors associated with snow rheology, in particular apparent cohesion, should also be taken into account. Considering a cohesion term is particularly relevant for modeling wet avalanche rheologies (Naaim et al., 2016; Bartelt et al., 2015a, b; Ligneau et al., 2022), whose proportion tends to increase in the context of climate change (Martin et al., 2001; Lazar and Williams, 2008; Castebrunet et al., 2014; Naaim et al., 2016). The presence of liquid water in snow leads to specific flow mechanisms and specific deposition patterns, different from those observed in dry flows (Fig. 1). Including cohesion in the classical Voellmy law is a first but crucial step to capture these behaviours (Bartelt et al., 2015a; Sanz-Ramos et al., 2023), which in turn affects the stoppage of avalanches and the transitions from *yielded* to *static* states.

In this context, the objective of this study is to address two main scientific questions, both related to the influence of numerical diffusion in avalanche simulations. First, is it possible to objectively define flow *arrest* on complex topographies, complementing a physically-based *yielding* criterion? Second, how do mesh resolution, terrain complexity, and flow cohe-

Natural Hazards and Earth System Sciences
EGU
Discussions

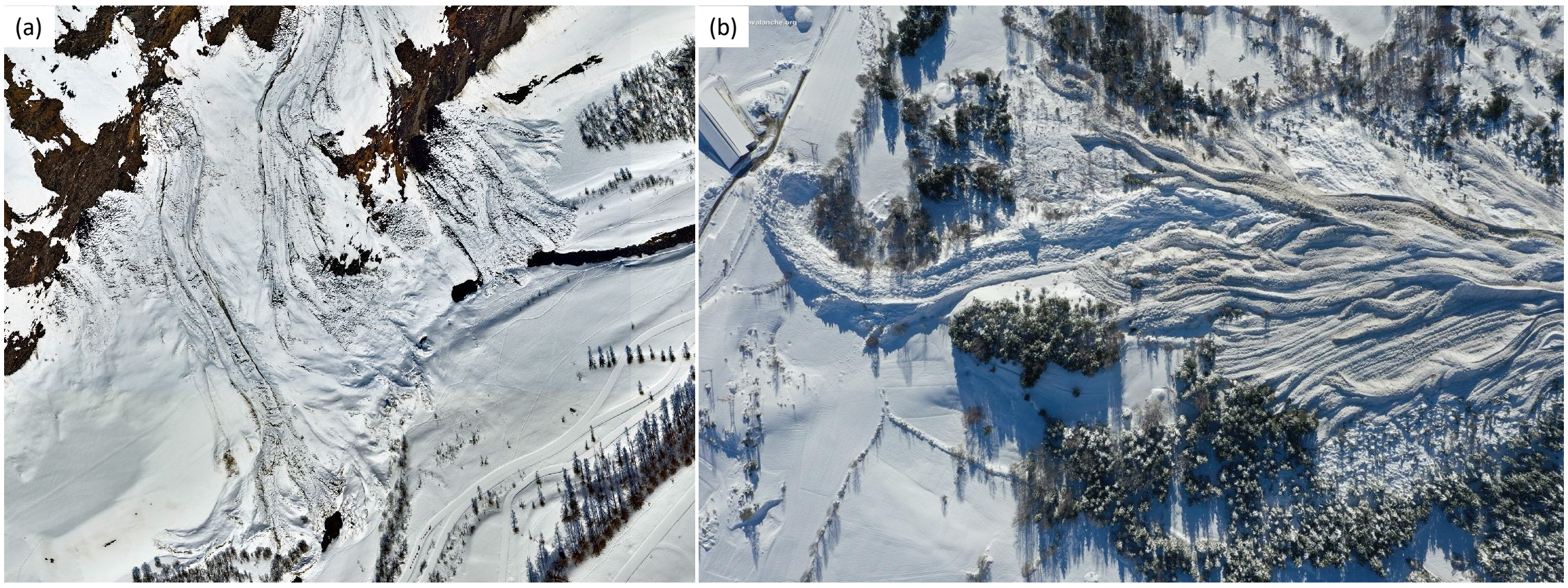

**Figure 1.** Two examples of wet snow avalanche deposits, with levees and fingering patterns: (a) Bessans area, Savoie, France (source: INRAE); (b) Termignon area, Savoie, France (source: www.data-avalanche.org - G. Courbat).

sion influence the *yielding* and *arrest* criteria? For that purpose, we use a well-validated second-order depth-averaged FVM model (Naaim et al., 2003, 2004, 2010; Chambon and Naaim, 2010), wherein we implement a specific *yielding* criterion and an enriched Voellmy law considering cohesion. Systematic sensitivity are performed in reduced-scale cases with no entrainment, to minimise confounding factors.

We firstly discuss the newly modified depth-averaged FVM model with a Voellmy friction law (§2), including the *yielding* criterion and cohesion $\tau_c$. We then present results where we vary the mesh resolution, terrain complexity, and $\tau_c$, to evaluate the new *yielding* criterion (§3). We then test several alternative post-processing *arrest* criteria, both globally (§4.1) and locally (§4.2). Finally, we compare our findings with those from other studies (§5).

## 2 Depth-averaged modeling of snow avalanches

### 2.1 Shallow-flow equations

We consider a local reference frame attached to the topography, with $x$ and $y$ the slope-parallel coordinates and $z$ the slope-normal coordinate. The depth-averaged equations expressing mass and momentum conservation under the shallow-flow assumption are:

$$\frac{\partial h}{\partial t} + \frac{\partial (hU)}{\partial x} + \frac{\partial (hV)}{\partial y} = 0, \tag{1}$$

$$\rho \frac{\partial (hU)}{\partial t} + \rho \left( \frac{\partial (hU^2)}{\partial x} + \frac{\partial (hUV)}{\partial y} \right) = \rho g_x h - \rho g_z h \left( \frac{\partial h}{\partial x} \right) - \tau_{zx}, \tag{2}$$

$$\rho \frac{\partial (hV)}{\partial t} + \rho \left( \frac{\partial (hUV)}{\partial x} + \frac{\partial (hV^2)}{\partial y} \right) = \rho g_y h - \rho g_z h \left( \frac{\partial h}{\partial y} \right) - \tau_{zy}, \tag{3}$$

Natural Hazards and Earth System Sciences
EGU
Discussions

where $h$ is the flow depth; $U$ and $V$ are the (slope-parallel) components of the depth-averaged velocity; $\rho$ is density (assumed constant throughout); $g$ is gravitational acceleration, considering slope-normal ($g_z$) and slope-parallel ($g_x$, $g_y$) components; and ($\tau_{zx}$, $\tau_{zy}$) are lumped basal resisting shear forces.

For snow avalanche modeling, resisting shear forces are generally expressed as the sum of a dry and a turbulent friction term, through the classical Voellmy model (e.g. Voellmy, 1955; Salm, 1993; Christen et al., 2010b; Naaim et al., 2013). We consider here an enriched Voellmy model, in which an apparent cohesion term $\tau_c$ is also included:

$$\tau_{zx} = \left[\rho g_z h\mu + \rho g \frac{U^2+V^2}{\xi} + \tau_c\right] \frac{U}{\sqrt{U^2+V^2}}, \quad (4)$$

$$\tau_{zy} = \left[\rho g_z h\mu + \rho g \frac{U^2+V^2}{\xi} + \tau_c\right] \frac{V}{\sqrt{U^2+V^2}}, \quad (5)$$

where $\mu$ is the dry (Coulomb) friction coefficient, and $\xi$ is the "turbulent" friction coefficient (see Salm, 1993). For snow avalanches, typical values of these coefficients are $\mu = 0.15 - 0.50$, and $\xi = 500 - 2000\ \mathrm{m.s^{-2}}$ (Salm et al., 1990). Values for cohesion $\tau_c$ remain poorly constrained and depend on snow characteristics including liquid water content and density. From large-scale snow chute experiments, Bartelt et al. (2015b) estimated cohesion values in the range 0–0.4 kPa for dry snow, and 0–2.3 kPa for wet snow; consistently, back-analyses of a real wet snow avalanche with a depth-averaged model led to a cohesion of about 1 kPa.

Note that Eqs. (2) and (3) disregard terms related to topographic curvature (e.g. Naaim et al., 2004; Fischer et al., 2012). These terms can induce numerical instabilities near small-scale topographical features, and can be considered as implicitly accounted for by the turbulent friction term when using Voellmy friction law (Peruzzetto et al., 2021). Equations (1)–(3) coupled to the Voellmy law enriched with cohesion (4)–(5) are solved numerically using an explicit second-order Godunov–Van Leer finite volume scheme (Vila, 1986; LeVeque, 2002). Spatial discretisation is based on a structured Cartesian mesh with square elements.

### 2.2 Yielding criterion

Physically, Eqs. (4)–(5) apply only for flowing material. An additional condition – a *yielding* criterion – should be added to describe that the material remains static when the local stress state lies below a flow threshold defined by cohesion and dry friction. In practice, we define $\tau_{b,\ \mathrm{test}}$ as the basal stress computed from momentum conservation, considering inertia, gravity and pressure gradient terms. If $\tau_{b,\ \mathrm{test}}$ lies under the flow threshold, the local velocity is set to zero, namely:

$$|\tau_{b,\mathrm{test}}| < \tau_c + \rho g_z h\mu \implies (U,V) = 0. \quad (6)$$

Condition (6) enables an objective identification of flowing (i.e. *yielded*) and static (i.e. *unyielded*) cells through a physically-based criterion. Note that imposing $U = 0$ and $V = 0$ does not "freeze" other flow parameters, e.g. static points can still change in height, given mass fluxes from non-static neighbouring points. Such mass fluxes can also put static points back into motion.

### 2.3 Terrain generation

The "simple topography" used in this study was inspired by a synthetic reduced-scale terrain from Wang et al. (2004). It includes a $45°$ slope and a horizontal base (the "runout zone"), as depicted in Fig. 2a. The domain is 40 m long and 20 m wide. For the "simple topography", both the slope and the horizontal base are flat and smoothly joined by a spline function. For the "complex topography", more representative of real avalanche paths, additional features are superimposed on this base terrain:

1. A uniformly distributed random noise with a magnitude of 0.01 m is added to each node, simulating small-scale terrain features, e.g. rocks.
2. Perlin noise (Perlin, 1985) is applied across the entire domain, using PERLIN_NOISE.PY Python implementation.

Perlin noise is a procedural algorithm used for generating realistic terrain morphologies. Inputs include frequencies, amplitudes, and seeds. Varying frequencies and amplitudes can generate terrain features of different spatial scales and relative magnitude, such as gentle undulations, clearly defined channels, and rocky outcrops. Changing the seeds affects the spatial distribution of the surface features, while keeping them similarly proportioned. For this study, one single combination of input parameters was chosen, resulting in a specific complex topography (see Appendix A).

Fig. 2 shows four examples of topographies so generated, for two different mesh resolutions. The histograms depict local inclination distributions. Considering the histograms for pairs of cases (c) and (d) indicates that the topographical features generated through Perlin noise are statistically comparable across different resolutions, which is a specificity of this method.

### 2.4 Input parameters and non-dimensional numbers

For all simulations shown below, we set $\rho = 300$ kg.m$^{-3}$, $\mu = 0.15$ and $\xi = 1000$ m.s$^{-2}$. Cohesion was varied in the range $1 \leq \tau_c \leq 100$ Pa. The avalanche initiation zone was cylindrical, with depth $h_0 = 0.6$ m and radius $r_0 = 1.8$ m. (For reference, the maximum flow spreading width is about 15 m, whilst the maximum downstream deposit depth is about 0.2 m.) The timestep was fixed at $10^{-4}$ s, based on a CFL condition. Tables in Appendix A summarise the parameter values adopted for the different simulations.

The Froude number $Fr$ is defined as

$$Fr = \sqrt{\frac{U^2 + V^2}{g_z h}} \tag{7}$$

In our simulations, $Fr$ varied in the range $0 - 15$, which is representative of real snow avalanches (see e.g. Sovilla et al., 2008; Köhler et al., 2018). We also define the dimensionless ratio between cohesive and gravitational stresses:

$$N_c = \frac{\tau_c}{\rho g_z h}. \tag{8}$$

The maximum value reached in our simulations was $N_c \approx 0.15$, for typical heights $h \approx 0.2$ m in the runout zone. Note that such values of $N_c$ would correspond to typical cohesion values in the range 0.5–2.5 kPa at full scale (for $1 < h < 5$ m), consistent with recent studies on wet snow (e.g Bartelt et al., 2015b; Kyburz et al., 2020) (see also previous discussion in section 2.1). The number $N_c$ exemplifies the inherent scale-dependence of the cohesion parameter $\tau_c$.

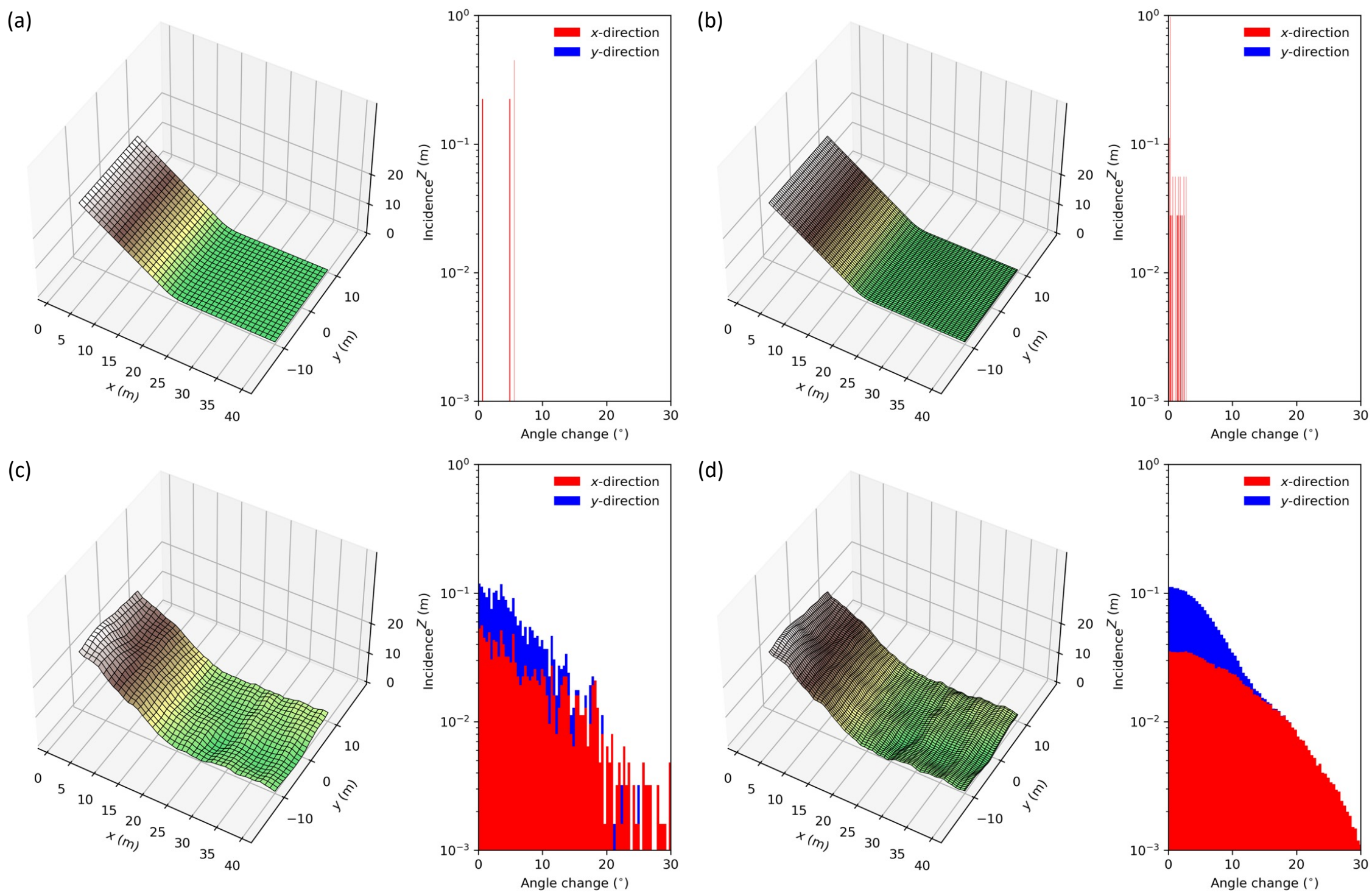

**Figure 2.** Examples of topographies generated in this study. (a) “Simple” topography with 40 × 20 cells; (b) same with 640 × 320 cells; (c) Complex topography with 40 × 20 cells; (d) same with 640 × 320 cells. The distributions of local slope angle variations between neighbouring nodes are also shown, determined using a 2 m sampling region.

## 3 Results: simulation output

### 3.1 Influence of mesh resolution

Figs. 3a and 3b display the model sensitivity to the grid resolution for the simple topography, showing time-histories of the positions of the centre of mass (CoM) and the frontal tip of the flow, respectively. Note that in general, the CoM position plateaus, indicating substantive flow stoppage, starting at times $4 < t < 8$ s. It is observed that grid sizes of 320×160 and coarser significantly influence simulation results for both $\tau_c = 1$ and 100 Pa, tending to slow flow propagation compared to higher-resolution simulations. Contrastingly, results converge for grid sizes of 640×320 or finer, and the beginning of the plateau (independent of the grid size) is then around $5$ s for $\tau_c = 100$ Pa and $7-8$ s for $\tau_c = 1$ Pa. We hence adopted a resolution of 640×320 for most simulations presented herein, i.e. around 10 cells per metre. Appendix B presents companion results obtained at lower resolutions, showing (for instance) that lower resolutions cause wider spreading of the flow.

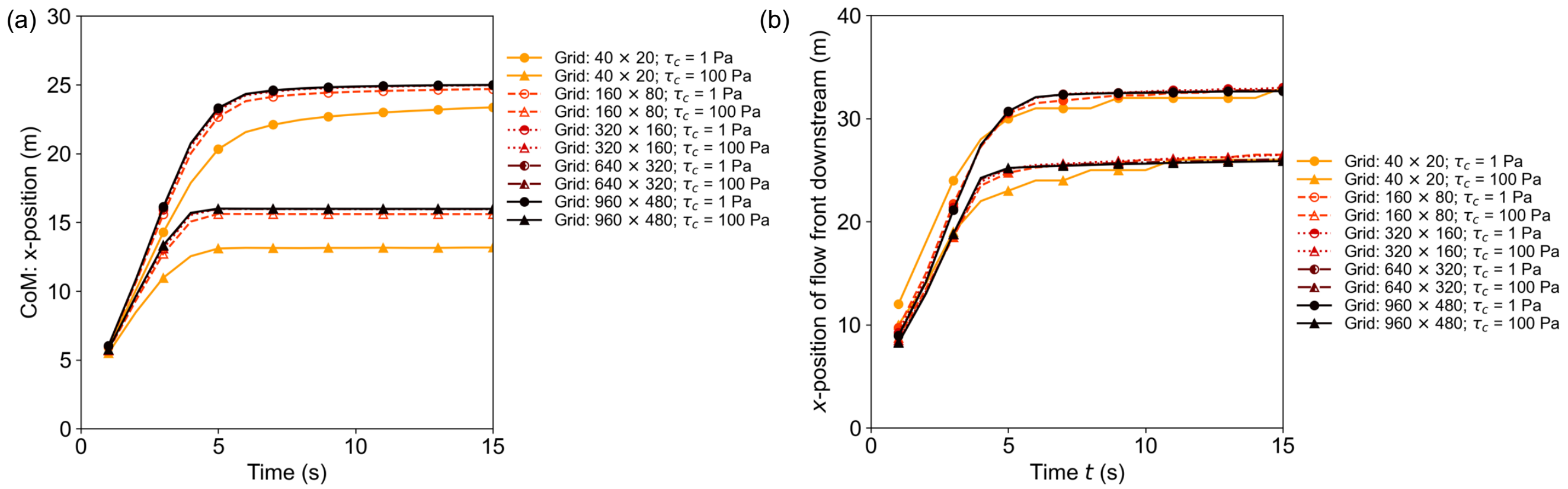

**Figure 3.** Position of (a) the centre of mass (CoM), and (b) the frontmost point of the flow, as a function of time, for the simple topography and different mesh resolutions. The marker types, circles or triangles, correspond to $\tau_c = 1$ Pa and 100 Pa, respectively.

## 3.2 Avalanche arrest mechanisms

### 3.2.1 Flow kinematics

Figs. 4a to 4c show snapshots, at different times, of avalanche propagation on the simple topography for $1 \leq \tau_c \leq 100$ Pa. For low cohesion values ($\tau_c = 1$ Pa, Fig. 4a), the flow stops from the bottom up. In contrast, for large cohesion values ($\tau_c = 100$ Pa, Fig. 4c), the flow stops from the top down, as cohesive stresses are sufficient to cause stoppage on the slope. Furthermore, lateral flow spreading is reduced relative to the case $\tau_c = 1$ Pa. For the intermediate value $\tau_c = 10$ Pa (Fig. 4b), the two arrest mechanisms (top-down, bottom-up) occur concurrently. Note that for all three cases, the limits of the yielded and static zones are very well defined, irrespective of the cohesion. Appendix C clarifies the breakdown of the different stress contributions leading to these arrest mechanisms.

Figs. 4d to 4f show corresponding results for complex terrains. As expected, the deposit morphology reflects the underlying topography. Although the effects of $\tau_c$ on the stopping mechanisms are similar to simple topographies, the limits of yielded and unyielded zones are much more "chaotic". For $\tau_c = 1$ Pa (Fig. 4d), a much smaller proportion of downstream material is static at $t = 8$ s compared to the simple topography case. This is due to local high inclinations causing residual flow even at large times. For $\tau_c = 100$ Pa (Fig. 4f), however, the proportion of static material is comparable at all timesteps to that obtained on the simple topography. For such high cohesion values, cohesive stresses are sufficient to cause full stoppage even on zones with locally high inclinations.

### 3.2.2 Proportion of *static* material

Fig. 5 shows the temporal evolution in *static* (unyielded) material for the same cases as in Fig. 4. For $\tau_c = 1$ Pa and a simple topography, the proportion of *static* material rapidly increases at $t \approx 5$ s, eventually plateauing at about 90% for $t \approx 8$ s. This

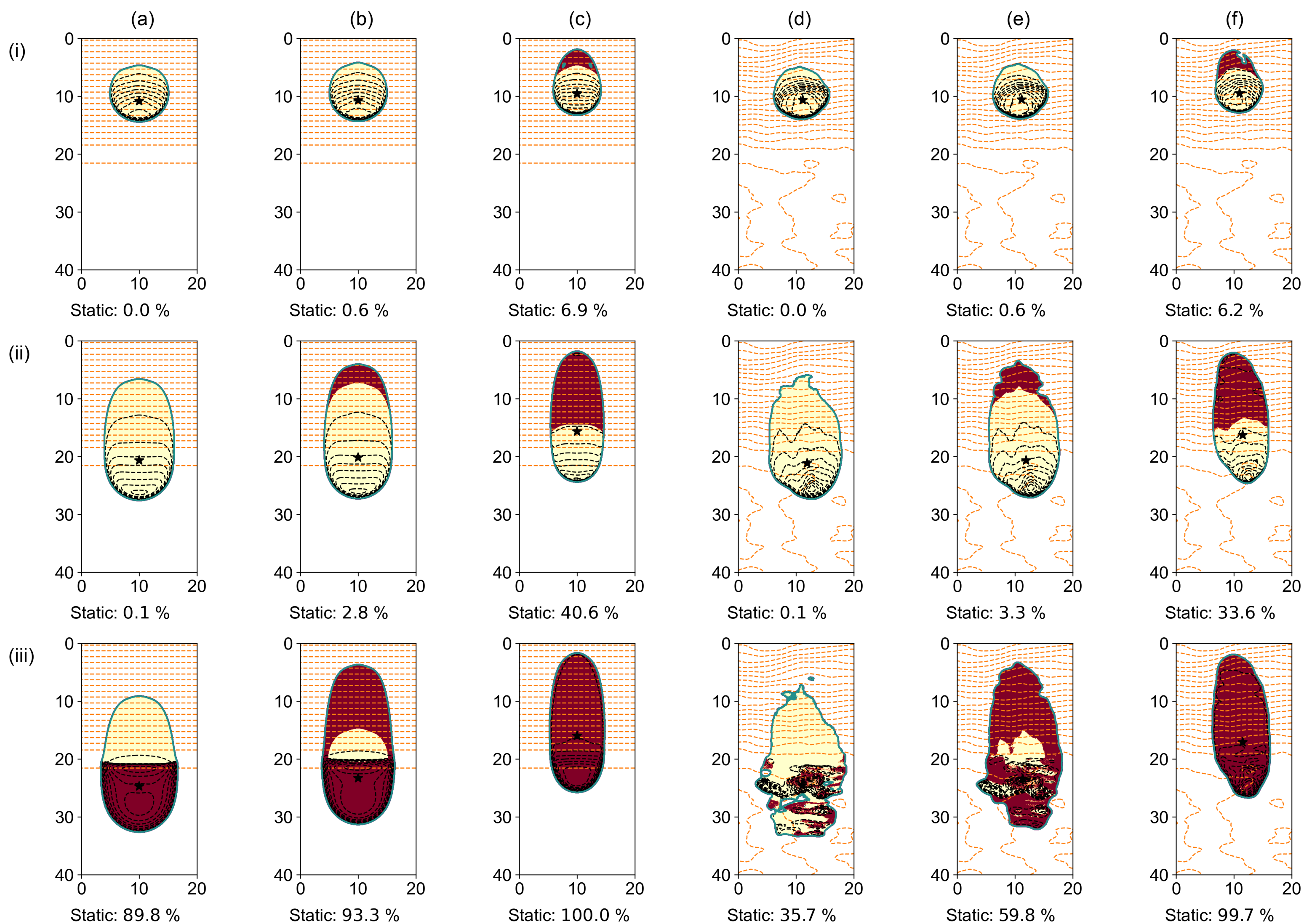

**Figure 4.** Flow snapshots for simple and complex topographies (grid size: 640×320). The $x$- and $y$-axes are in m. (a) $\tau_c$ = 1 Pa; simple topography; (b) $\tau_c$ = 10 Pa; simple topography; (c) $\tau_c$ = 100 Pa; simple topography; (d) $\tau_c$ = 1 Pa; complex topography; (e) $\tau_c$ = 10 Pa; complex topography. (f) $\tau_c$ = 10 Pa; complex topography. Parts (*i*), (*ii*) and (*iii*) correspond to $t = 2, 4$ and 8 s, respectively. Yellow regions are flowing, whilst red regions are *static* (from the yielding criterion). Dashed lines correspond to iso-depth contours.

Natural Hazards and Earth System Sciences
EGU
Discussions

corresponds to the stoppage of the flow when it reaches the horizontal runout zone. The remaining 10% correspond to material still slowly flowing downslope. For $\tau_c = 10$ Pa, the trend is similar, albeit with a slightly higher plateau. For $\tau_c$ = 100 Pa, the increase is more gradual, starting at $t \approx 1$ s, due to material depositing on the slope. The plateau is reached at $t \approx 5$ s, with a value of almost 100% in this case. Note that these trends, and in particular the plateauing times, are consistent with the evolutions of the CoM and frontal tip shown in Fig. 3.

Returning to Fig. 5, on the complex topography, for both $\tau_c = 1$ and 10 Pa, the proportion of static material rises more slowly than on the simple topography. Furthermore, transient maxima are observed due to the re-mobilisation of static material in the runout zone by the material flowing downslope. The final proportion of static material is notably lower than for the simple topography, being $\approx 40\%$ and 60% for $\tau_c = 1$ Pa and 10 Pa, respectively. For these cases, although a final steady state appears to be reached, the assessment of flow stoppage based on the proportion of static material would rely on a subjective threshold. This implies that sole reliance on the yielding criterion for defining avalanche arrest is insufficient on the complex topography when cohesion is low. In contrast, for $\tau_c = 100$ Pa, the topographical complexity only weakly influences the proportion of static material. A full stoppage is attained at $\approx 5$ s, similar to the simple topography. For large enough cohesion values, this indicator thus unambiguously pinpoints the time at which a fully static state has been reached, and is sufficient to objectively define avalanche arrest.

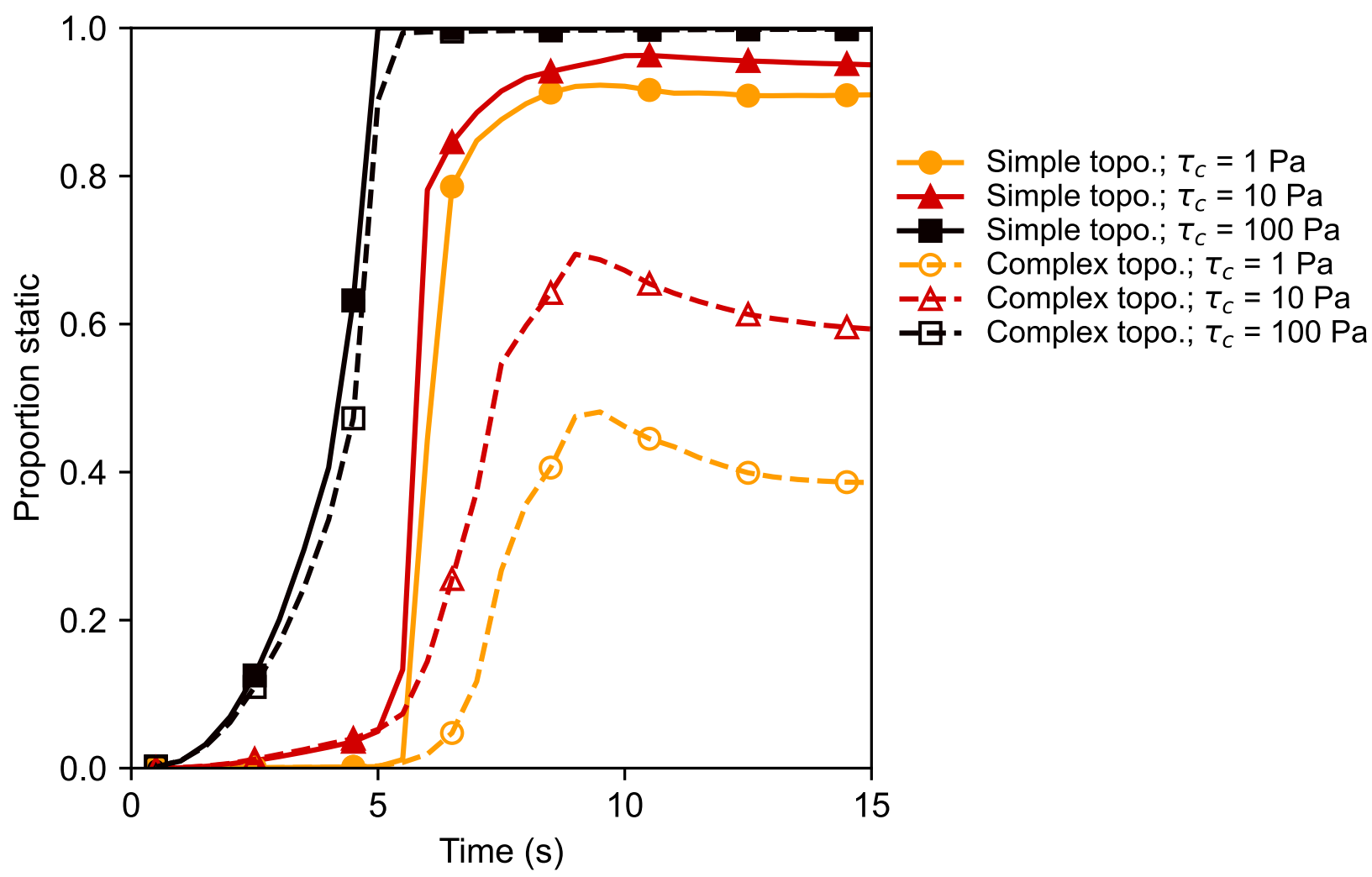

**Figure 5.** Proportion of *static* material (by volume) as a function of time. Results for different values of cohesion $\tau_c$, and for simple and complex topographies, are shown.

Natural Hazards and Earth System Sciences
EGU
Discussions

#### 3.2.3 Velocity distribution

Fig. 6 shows histograms of local $x$-velocities for the same cases as in Fig. 4. Each plot shows results for simple (yellow) and complex (black) topographies. The percentage of static material (excluded in the histograms) is indicated in the legends. Appendix D presents the same data plotted as a function of the underlying terrain inclination, to highlight the combined effects of Coulomb friction $\mu$ and cohesion $\tau_c$ on flow dynamics.

At time $t = 2$ s, is it observed that cohesion $\tau_c$ and the topographical complexity only weakly affect the results, with relatively uniform velocity distributions across the spectrum. However, note that the maximum velocities are slightly higher on the simple topography vs. the complex one; the maximum velocities tend to decrease as cohesion increases; and on the complex topography, numerous points are characterised by a near-zero velocity (the black spike on each plot).

At $t = 8$ s and for $\tau_c = 1$ Pa (Fig. 6a(*iii*)), the shape of the velocity distributions are also relatively similar for simple and complex topographies, although the percentage of static (unyielded) material differs, being 89.9 and 59.8%, respectively. Note that, although the CoM has broadly stopped at this point (Fig. 3), some material still moves at significant velocities even on the simple topography. Similar trends are observed for $\tau_c = 10$ Pa (Fig. 6b(*iii*)), albeit with lower maximum velocities. For $\tau_c$ = 100 Pa, (Fig. 6c(*iii*)), almost 100% of the material has stopped for both topographies. Note that the spike at low velocities ($U < 0.2$ m/s) for the complex topography is observed for all timesteps, and appears at $t$ = 8 s for the simple topography. This peak presumably mainly corresponds to points that should be physically arrested and are affected by numerical diffusion, although objectively distinguishing between numerical diffusion and genuine physical movement during flow deceleration remains challenging.

The sensitivity of flow stoppage to cohesion and terrain complexity can be explained by two mechanisms: (*i*) for low $\tau_c$, the basal shear resisting forces can enable a static state only on shallow inclinations (governed by $\mu$; see also Appendix D); and (*ii*) lower values of $\tau_c$ cause larger flow spreading, and hence thinner deposits, so complex terrain features strongly influence the avalanche stoppage. Both mechanisms contribute to non-convergence of velocities to zero for lower values of $\tau_c$.

Natural Hazards and Earth System Sciences
EGU
Discussions

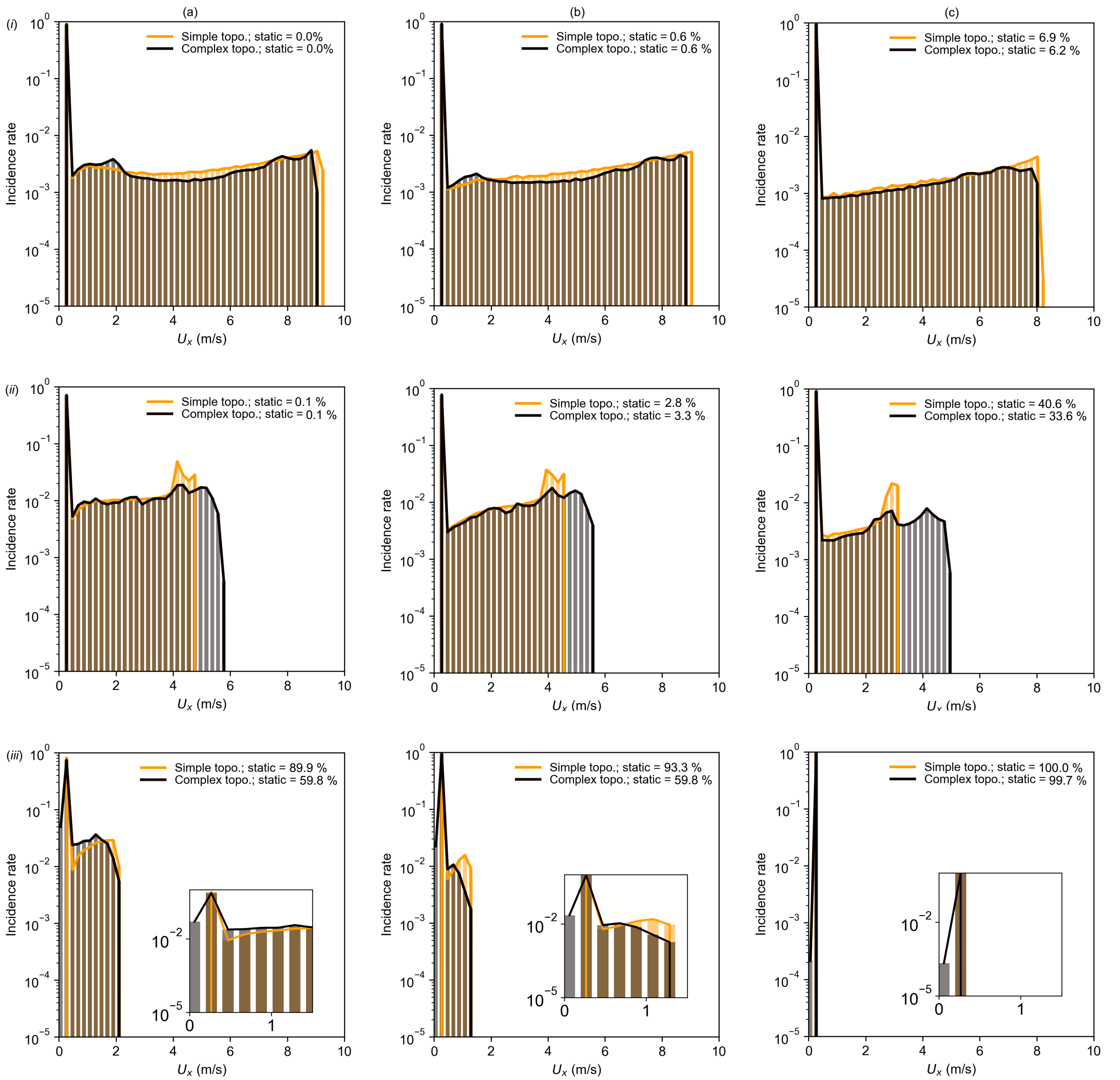

**Figure 6.** Velocity distributions (bin width: 0.2 m/s) for (a) $\tau_c$ = 1 Pa; (b) $\tau_c$ = 10 Pa; and (c) $\tau_c$ = 100 Pa, at (*i*) $t = 2$ s, (*ii*) $t = 4$ s and (*iii*) $t = 8$ s. Each histogram is normalised to unity, having discounted static points. The insets show zoomed-in versions of the main plots.

# 4 Results: *arrest* criteria

## 4.1 Global arrest criteria

As shown in the previous section, relying on the physically-based yielding criterion is generally insufficient for defining avalanche arrest objectively when a realistic, complex topography is considered. Complementary *arrest* criteria are thus required. Classically, such arrest criteria rely on arbitrarily defined velocity or momentum thresholds defined globally across the flow (see §1). In this section, we explore the capabilities of such global thresholds, focusing on indicators related to flow velocity.

### 4.1.1 Using averaged velocities

Fig. 7 shows the avalanche centre of mass (CoM) velocity $U_{\mathrm{CoM}}$ against time for several values of cohesion $\tau_c$. The velocity is derived from the CoM position (Fig. 3(a)) using a Savitzky–Golay smoothing filter with a window length of 5 points and a polynomial order of 3. For $t < 3$ s, where inertial effects generally prevail, terrain complexity and $\tau_c$ only minimally affect $U_{\mathrm{CoM}}$. For $\tau_c = 100$ Pa, however, inertial and cohesive stresses have similar orders of magnitude such that $U_{\mathrm{CoM}}$ assumes lower values even at early times. For $t > 3$ s and the simple topography, it is observed that higher $\tau_c$ causes the flow to decelerate sooner, as expected. The divergence between the curves for various values of $\tau_c$ occurs around $t = 5$ s, which roughly corresponds to the apparent stopping time of the CoM in Fig. 3. After this time $U_{\mathrm{CoM}}$ drops and reaches values below $10^{-2}$ m/s, typically. Note however that this velocity drop is much more rapid for $\tau_c > 10$ Pa, while $U_{\mathrm{CoM}}$ decreases more gradually for lower cohesion values. On the complex topography, the trend between $\tau_c$ and $U_{\mathrm{CoM}}$ for $t > 3$ s is no longer monotonic, as $U_{\mathrm{CoM}}$ is also affected by the specificities of the underlying terrain. For instance, the consistent dip in the velocities at $t \approx 6$ s, in particular for low values of $\tau_c$, corresponds to a short net backward motion of the avalanche due to local upward-facing slopes in the runout zone.

The different curves in Fig. 7 highlight the difficulty of defining a global velocity threshold that would delineate flow stoppage in all cases, in particular for low cohesion values. This difficulty is even more pronounced on a complex topography, due to local flow mechanisms (e.g. temporary backward movement) that can affect the average avalanche velocity. We note however that, both for simple and complex topographies, most of the curves display a change in the slope of the evolution of $U_{\mathrm{CoM}}$ over time for $t$ between 5 and 8 s, typically. This change presumably corresponds to the transition between actual physical movement and long-term evolution dominated by numerical diffusion. Potentially, it could thus offer a more objective criterion to define effective flow arrest, although it would still require to be manually pinpointed on the curves.

### 4.1.2 Setting a velocity threshold

Velocity-based thresholds can also be used to define *arrested* material within the flow, thus complementing the physically-based yielding criterion. Figs. 8a and b show the proportion of arrested material defined for two variations of such thresholds and two velocity cutoffs $U_{\mathrm{thresh}} = 1$ and $10$ cm·s$^{-1}$. These cutoff values were selected based on the results from Fig. 6 and

Natural Hazards and Earth System Sciences
EGU
Discussions

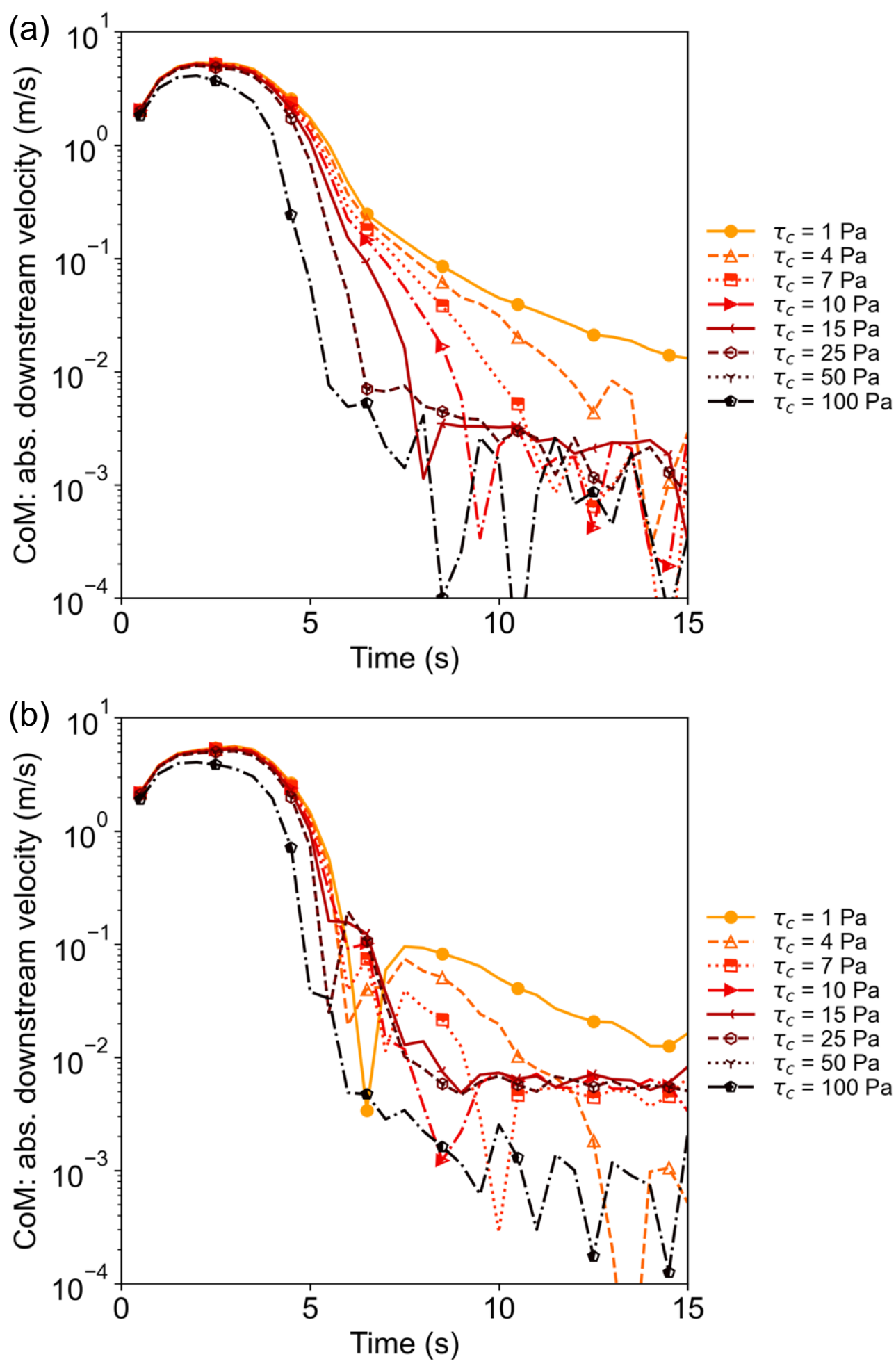

**Figure 7.** Velocity of the centre of mass of the flow as a function of time and for different values of cohesion $\tau_c$: (a) simple topography; (b) complex topography. Note that the complex topography used for all simulations shown here was identical; other topographical distributions give rise to variations in the velocity histories.

Fig. 7, as plausible representative values below which flow dynamics is likely strongly affected by numerical diffusion. In Fig. 8a, all points lying below these velocity cutoffs are classified as arrested. Fig. 8b additionally includes a topography-related criterion: the velocity cutoff is only applied where the local inclination angle $\theta$ is less than the friction angle computed from the dry friction coefficient $\mu$, namely $\theta < 9°$. The rationale for this additional criterion is that, for slopes shallower than this threshold, Coulomb friction will act to decelerate and ultimately stop the flow, were it not for numerical diffusion. Note that, due to the additional contribution of the cohesion $\tau_c$, this criterion is rather conservative; Appendix D elaborates further on the influence of $\tau_c$ on the distribution of velocities within the flow, considering the underlying topography inclination. The average local inclination for each cell is computed using a 2 m diameter window to discriminate between the slope and the flattish zones where material tends to deposit.

By construction, the proportion of arrested material defined according to the velocity-based criteria described above is larger than the proportion of static material defined based solely on the yielding criterion (Fig. 5). The overall trends, however, of the proportions of *arrested/static* material remain consistent. Fig. 8a($i$) shows that, with $U_{\text{thresh}} = 1$ cm·s$^{-1}$, the flows on the simple topography are mostly arrested at $t < 5$ s for $\tau_c = 100$ Pa, and at $t = 8-10$ s for lower cohesion values. For the complex topography, simulations with cohesion values of $\tau_c = 1$ and 10 Pa continue to show a significant proportion of flowing material even at $t = 15$ s. Increasing $U_{\text{thresh}}$ to 10 cm·s$^{-1}$ (Fig. 8a($ii$)) results in all flows being mostly arrested by a maximum time of around $t = 10$ s. In Fig. 8b, the proportion of arrested material is lower than in Fig. 8a for both $\tau_c = 1$ and 10 Pa, and both topography conditions. Due to the addition of the criterion related to slope angle, the material located on sloping parts of the topographies is here classified as flowing. Note that the results obtained for this criterion are very close to those provided by the yielding criterion only (compare with Fig. 5).

## 4.2 Locally-based arrest criterion

For simple and complex topographies, Figs. 9 and 10 show time-histories of different metrics relating to the highest point of the flow material on the runout zone: namely, the velocity $U_{\text{hp}}$, the height $h_{\text{hp}}$, the longitudinal position $x_{\text{hp}}$, and the flowing/static state (according to the yielding criterion) of this point. The runout zone is defined here as the zone for $x > 20$ m. Note that the highest point is defined at each timestep, and is not a Lagrangian quantity. Results for different mesh resolutions and two cohesion values $\tau_c = 1$ Pa and 100 Pa are shown.

For simple topography and $\tau_c = 1$ Pa (Fig. 9a), the velocity of the highest point $U_{\text{hp}}$ defaults to 0 at $t \approx 6$ s, independently of the resolution. After this time, this point is consistently in a static state according to the yielding criterion. Note however that the position $x_{\text{hp}}$ of this point continues to move downstream, and the height $h_{\text{hp}}$ to slowly decrease, presumably due to numerical diffusion. Increasing the resolution tends to reduce the decrease rate of $h_{\text{hp}}$, but the decrease remains perceptible even for the larger resolution values. For $\tau_c = 100$ Pa (Fig. 9b), the highest point becomes static sooner, at $t < 5$ s, consistent with previous results. Here, the rate of decrease $h_{\text{hp}}$ is smaller than for $\tau_c = 1$ Pa, and becomes essentially imperceptible for the larger resolutions, even if the values of $h_{\text{hp}}$ are similar in the two cases.

For the complex topography (Fig. 10), results are generally “noisier”, with several successive transitions between flowing and static states for the highest point, in particular for $\tau_c = 1$ Pa. The height $h_{\text{hp}}$ displays more pronounced changes than for the

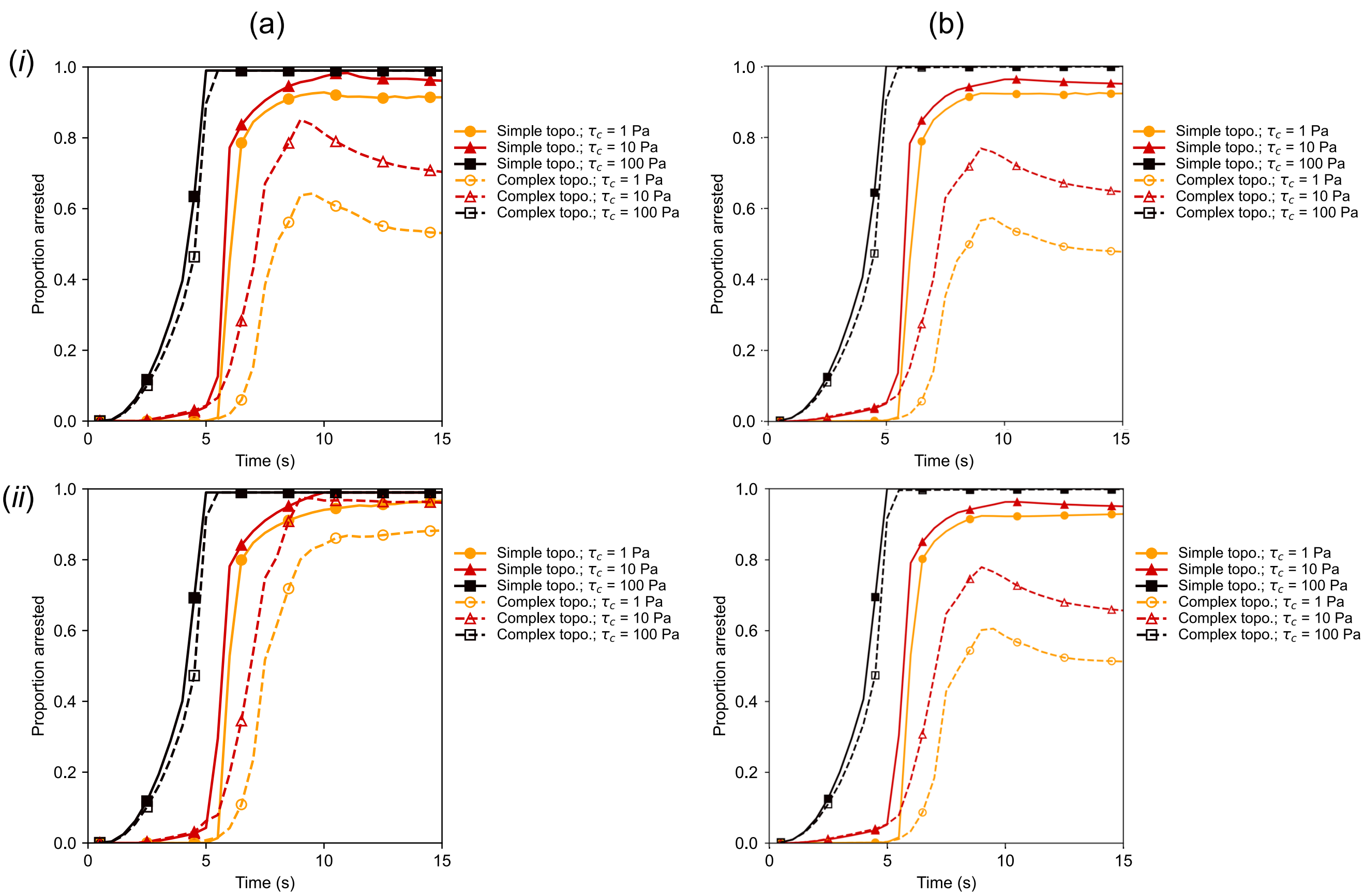

**Figure 8.** Proportion of *arrested* material (by volume) defined from velocity-based criteria, as a function of time. (a) Arrest criterion based on velocity thresholds set at 1 and 10 cm $\cdot$s$^{-1}$ for parts (*i*) and (*ii*), respectively (i.e. any point with a velocity $< 1$ or $< 10$ cm$\cdot$s$^{-1}$ is defined as arrested.) (b) Same arrest criterion as in (a), but only for points where local terrain inclination $\theta < 9^\circ$ (see text). The default yielding criterion is used where $\theta \geq 9^\circ$.

simple topography, notably for $\tau_c = 100$ Pa and when the highest point is static. These long-term evolutions of $h_{\mathrm{hp}}$ are a clear hallmark of enhanced numerical diffusion on complex topography. Note that the larger values of $h_{\mathrm{hp}}$, compared to the simple topography, are related to the different runout distances and hence to the particularities of the underlying topography. Yet, for $\tau_c = 100$ Pa, the velocity $U_{\mathrm{hp}}$ consistently drops to 0 at $t \approx 5.0$ s across all resolutions (Fig. 10b). This time corresponds to the first moving-to-static transition of the highest point. For $\tau_c = 1$ Pa, the first moving-to-static transition occurs at $t \approx 7.5$ s (Fig. 10a). The velocity $U_{\mathrm{hp}}$ remains also essentially null after this time, although a small residual noise can be observed, presumably attributable to the enhancement of numerical diffusion by the topographical complexity.

Interestingly, particular metrics related to the highest point in the runout zone, notably the first transition to the static state of this point and the cancellation of its velocity, appear to offer objective solutions for defining avalanche arrest. Arrest criteria

Natural Hazards and Earth System Sciences
EGU
Discussions

based on these metrics are relatively unaffected by numerical diffusion, and remain effective even for complex topography, low cohesions, and/or low resolution values.

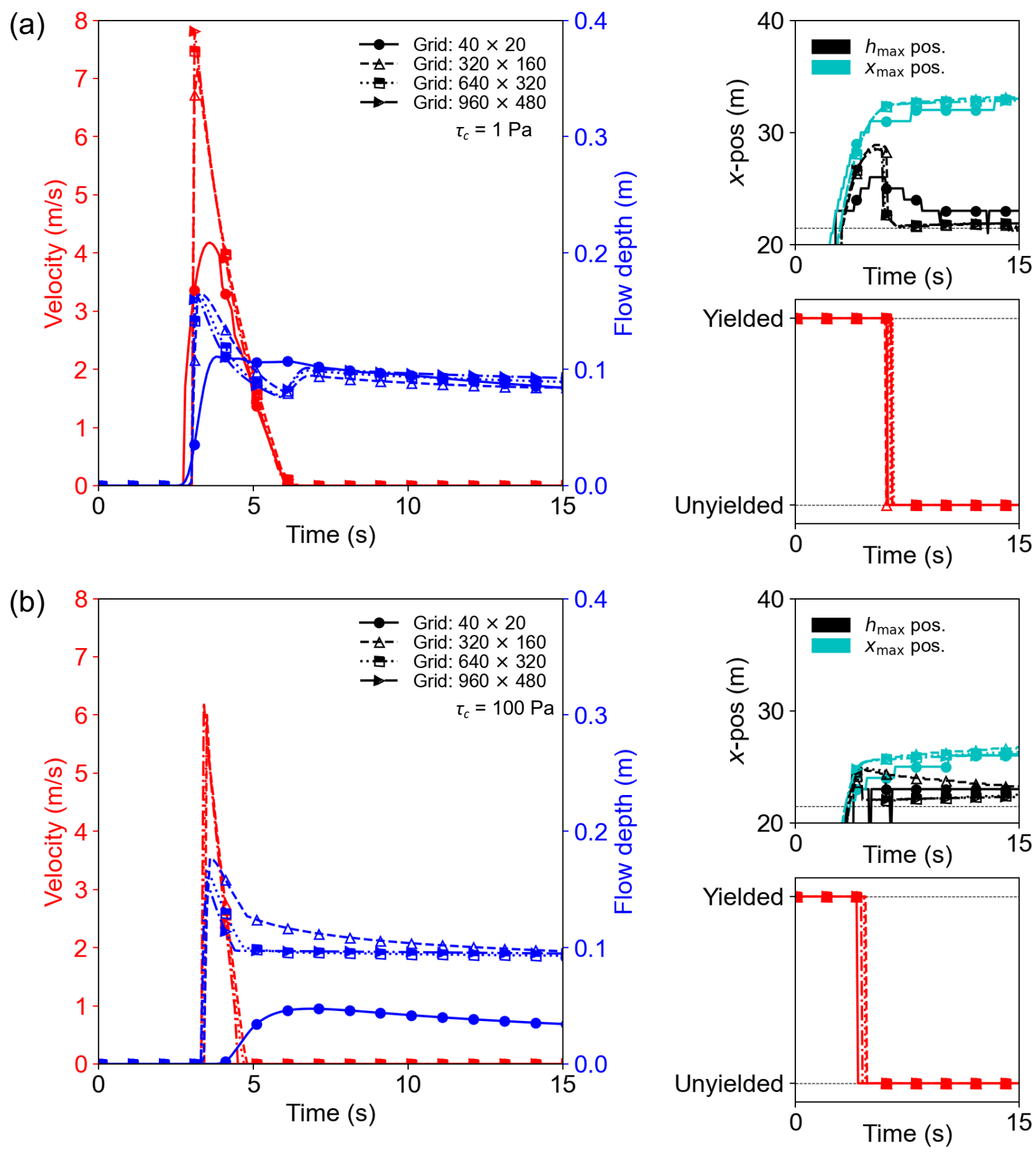

**Figure 9.** Metrics relating to the highest point (hp) in the downstream flow material as a function of time, for simple topography and different mesh resolutions. (a) $\tau_c$ = 1 Pa; (b) $\tau_c$ = 100 Pa. Clockwise, the subplots show: the velocity and height of hp; the longitudinal position of hp, as well as the position of the avalanche front tip (downstream-most flow point); moving or static state of hp from the yielding criterion.

## 5 Discussion

Modeling avalanche propagation with the help of depth-averaged models usually requires complementary arrest criteria for defining flow stoppage, especially on realistic, complex topography. However, sufficiently robust and objective arrest criteria are hitherto lacking. This manuscript considers the interplay between three key variables on avalanche arrest, namely mesh resolution, topography complexity, and snow cohesion. Our simulation model includes a physically-based yielding criterion

Natural Hazards and Earth System Sciences
EGU
Discussions

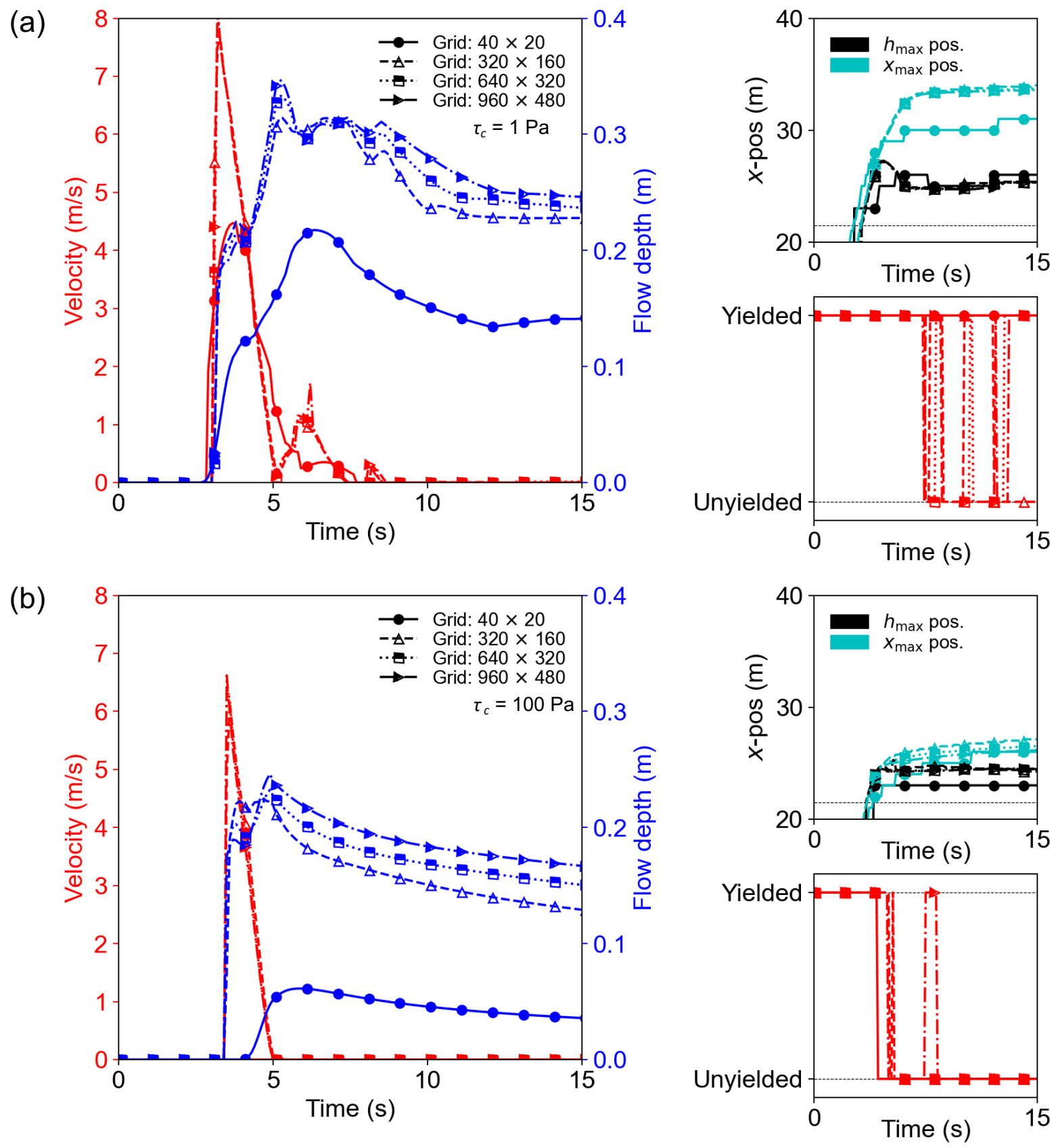

**Figure 10.** Metrics relating to the highest point in the downstream flow material as a function of time, for complex topography and different mesh resolutions. Same legend as Fig. 9.

based on frictional and cohesive stresses. Within this framework, we trialed the applicability of various arrest criteria, and their capability to differentiate between physical flow and numerical diffusion when simulated avalanches are close to stoppage.

Only a few avalanche simulation models implement a physically-based yielding criterion to identify zones in which stresses lie below the flow threshold of the material (Zugliani and Rosatti, 2021; Sanz-Ramos et al., 2023). Yet, we found that robustly defining flow arrest on the sole base of such a yielding criterion is only possible for cases with sufficiently high resolutions (640×320 or higher) and high cohesion values ($\tau_c = 100$ Pa in our simulations). In these configurations, the proportion of static material reaches almost 100% at flow stoppage. For all other cases, a significant proportion of material remains yielded even after long times, due to a combination of residual physical flow in locally inclined areas (on complex topography) and numerical diffusion. Although the typical numbers indicated above may vary, we can expect this consequence of numerical diffusion to be largely independent of the numerical scheme and the type of yielding criterion implemented. In particular, it

would be interesting to conduct similar systematic sensitivity analyses with other models (Zugliani and Rosatti, 2021; Sanz-Ramos et al., 2023) to get a broader overview of the limitations of yielding criteria on complex topography.

In general, a specific arrest criterion is thus required, even when a yielding criterion is implemented. In practice, setting up arrest criteria on the base of global thresholds proved difficult. The evolution of average flow velocity, and the "final" residual value attained at rest, is highly variable as a function of topography complexity and cohesion (Fig. 6). Although the transition between physical flow and the regime dominated by numerical diffusion is visible in the curves, the automatic detection of this transition is challenged by the complexity of arrest mechanisms, notably on complex topography. The detailed investigation of velocity distributions can also suggest relevant velocity thresholds, below which the flow is mainly governed by numerical diffusion (Fig. 7). Such thresholds can be used to complement the yielding criterion and define zones in which the material is effectively arrested, even if it has not reached the static state yet. However, in general, adopting such velocity thresholds does not fundamentally alter the evolution of the proportion of static/arrested material, nor the fact that the final proportion of static/arrested never reaches 100% (unless unrealistically high threshold values are used). The time to reach the peak or plateau appears largely unchanged (compare Fig. 5 to Fig. 8). This time can be used as a proxy to define avalanche arrest when conducting parametric studies on particular topographies. Again, however, the various behaviours observed when varying topography and cohesion would make it difficult to base an objective arrest criterion on this time. Note that, in this study, we limited consideration to velocity-based thresholds. We do expect, however, the above observations to remain essentially identical for other global arrest thresholds based on, e.g., flow momentum or energy.

In this work, we also explored a new "local" arrest criterion considering a representative point in the avalanche, specifically the highest point in the runout zone. For a general topography, this zone can be defined as the zone where $\theta < \mu$ over a suitably large extension. This local approach avoids having to "wait for" all flow points to become static. Detecting the first transition of this representative point to the static state, using the physically-based yielding criterion, obviates arbitrary velocity thresholds and seems to offer a promising and objective way to define flow arrest and terminate the simulation. Note that tracking this highest point is physically reasonable since this point is expected to be the most prone to moving due to large driving stresses. Note also that the evolution of the height of this point $h_{\mathrm{hp}}$ exemplifies the effect of numerical diffusion, in that $h_{\mathrm{hp}}$ can continue to change even when the point is static.

Interestingly, we found that mesh resolution has only little effect on the first transition of this highest point to the static state, or on its velocity (Figs. 9 and 10). Accordingly, the local arrest criterion would remain largely independent of resolution, at least for the cases investigated in this study, which appears as a significant asset. This contrasts with other results from this study, which indicate that global arrest criteria would on the contrary be strongly dependent on mesh resolution. In particular, lower resolutions tend to slow down flow propagation and increase flow spreading (compare also Figs. 4 and B1). Similarly, other numerical studies considering real full-scale topographies (Christen et al., 2010a; Bühler et al., 2011; Miller et al., 2022) reported that avalanche runout distances, defined on the base of a global criterion, are prone to be affected by mesh resolution.

Finally, it should also be underlined that increasing the value of cohesion tends to facilitate the definition of avalanche arrest. Above a certain value, cohesion can "freeze" the flow on sloping terrains, thereby counteracting the effects of numerical diffusion. In this case, tracking the proportion of arrested material, following either the yielding criterion or velocity-based

criteria, can offer a sufficient and practical arrest criterion both on simple and complex topographies (see Figs. 5 and 8 for $\tau_c = 100$ Pa). In practice, enriching the Voellmy law with a cohesive term could thus be recommended whenever an unambiguous definition of avalanche arrest is of importance. One can wonder, however, whether cohesion values typical of dry snow would be sufficient to activate this effect in real-scale avalanches. As an alternative, increasing cohesion in the runout zone, or below a specific velocity threshold, could be an option. Complementary analyses on real-scale test cases would be needed to further explore this issue.

## 6 Conclusions and future work

We simulated snow avalanches over synthetic terrains using a depth-averaged flow model. The model is based on a cohesive Voellmy law, including a physically-based yielding criterion to detect zones where the material is static. We subsequently investigated the influence of mesh resolution, topographical roughness, and snow cohesion $\tau_c$ on the avalanche stoppage mechanisms, and the possibility to define objective *arrest* criteria.

We found that mesh resolution affects simulated flow properties if the cell size is larger than about 20% of the characteristic flow depth, typically, both for simple and complex topographies. In particular, the final proportion of static material decreases and the extent of flow spreading increases with coarser mesh resolution, through enhanced numerical diffusion. However, resolution exerts less influence on the depth of the deposit in the runout zone.

Large cohesion values were observed to promote a full transition of the material to a static state even on complex and steep slopes ($> 45°$). Cohesion limits both lateral and longitudinal flow spreading and increases flow thickness. Consequently, large cohesion values reduce the influence of topographical roughness on the final deposit. Conversely, avalanches with low cohesion values are generally thinner and more influenced by terrain features in their stopping phase. Locally sloping zones induce non-zero residual velocities and slow down the transition to a fully static state.

Our investigation also revealed that arrest criteria relying on whether local velocities across the avalanche lie below a threshold are applicable for high cohesion values, as most points eventually become static. For low cohesion, however, this approach requires setting ad-hoc thresholds in the proportion of arrested material, due to the persistence of numerous points with non-zero velocities. Similarly, defining an objective arrest criterion based on flow averages (e.g. the velocity of the centre of mass) is challenged by the complexity of flow mechanisms and the gradual transition from physical flow to artificial numerical diffusion. A promising objective arrest criterion considering the transition to a static state of a particular, representative point in the flow was also proposed. We selected here the deepest point of the downstream flow material. This criterion appears applicable for both simple and complex topographies across all tested cohesion values. It also appears relatively insensitive to mesh resolution.

Identifying when/whether avalanches are arrested is of paramount importance for, e.g., hazard zoning or designing mitigation measures. This study demonstrates that systematic sensitivity analyses on the chosen arrest criteria are essential as soon as numerical models are to be relied upon for such operational applications. The influence of numerical diffusion is expected to be dependent on the specificities of the numerical scheme and, as such, the interplay between topography, rheology and mesh

resolution might vary among the different existing avalanche flow models. Note that, since it does not rely on any subjective threshold, the proposed arrest criterion based on tracking a specific point in the avalanche could offer a way to demarcate flow arrest independently of the numerical scheme. The implementation of a yielding criterion, however, appears essential for the applicability of this approach.

Finally, let us recall that this study only considered simple avalanches on model topographies. Similar sensitivity analyses on real full-scale topographies will be needed to further ascertain the conclusions and, in particular, validate the applicability of the proposed local arrest criterion. In real cases, processes such as splitting of the flow into several branches, or impacts against counter-slopes, may further complicate the detection of avalanche arrest. Along a similar line, the influence of the dry and turbulent friction coefficient, $\mu$ and $\xi$, which were here maintained constant, would need to be investigated in conjunction with variations in cohesion $\tau_c$. Note also that the typical value of $\tau_c$ above which flow arrest is dominated by cohesion effects, i.e. approx. 50 Pa in this study, is expected to be dependent on avalanche thickness and will need to be studied more systematically. Lastly, additional complexities in the arrest mechanisms may also arise from entrainment and deposition processes, the proper inclusion of which in shallow-flow models presently remains an open issue (Issler, 2014; Li et al., 2022).

**Acknowledgements**

This study was funded by the MOPGA 2021/22 scheme (MOPGA-976501H), the ANR, and “Direction Générale de la Prévention des Risques” from the French Ministry for Environment.

**Code/Data availability**

On request, data generated from this study will be made available by the authors. We are willing to host the data generated from the study in a public repository upon potential acceptance of the manuscript.

**Author contributions**

Saoirse Goodwin: Data curation; Formal analysis; funding acquisition; investigation; methdology; software; validation; visualisation; writing - original draft preparation. Thierry Faug and Guillaume Chambon: Conceptualisation; Formal analysis; Funding acquisition; Methodology; Project administration; Resources; software; supervision; validation; visualization; writing - review & editing.

**Competing interests**

The authors declare that there are no competing interests.

## Nomenclature

| | |
|---|---|
| $\mu$ | Dry Coulomb friction efficient [-] |
| $\rho$ | Density [kg/m$^3$] |
| $\tau_c$ | Apparent cohesion [Pa] |
| $\tau_{b,\text{test}}$ | Basal stress threshold [Pa] |
| $\tau_{zx}$ | Lumped basal-resisting shear forces (downstream) [Pa] |
| $\tau_{zy}$ | Lumped basal-resisting shear forces (cross-stream) [Pa] |
| $\theta$ | Local inclination of slope [°] |
| $\xi$ | "Turbulent" friction coefficient [m/s$^2$] |
| $Fr$ | Froude number [-] |
| $g$ | Acceleration due to the Earth's gravity |
| $g_x$ | Slope-parallel acceleration due to the Earth's gravity (cross-stream) |
| $g_x$ | Slope-parallel acceleration due to the Earth's gravity (downstream) |
| $g_z$ | Slope-normal acceleration due to the Earth's gravity |
| $h$ | Flow depth [m] |
| $h_{\text{hp}}$ | Depth of the highest point in the flow where average topography inclination is nominally less than basal friction angle [m] |
| $N_c$ | Dimensionless ratio between cohesive and gravitational stresses [-] |
| $t$ | Time [s] |
| $U$ | Slope-parallel depth-averaged velocity (downstream) [m/s] |
| $U_{\text{CoM}}$ | Velocity of the centre of mass (downstream) [m/s] |
| $U_{\text{hp}}$ | Velocity of the highest point in the flow where average topography inclination is nominally less than basal friction angle (downstream) [m/s] |
| $U_{\text{thresh}}$ | Velocity threshold (downstream) [m/s] |
| $V$ | Slope-parallel depth-averaged velocity (cross-stream) [m/s] |

Natural Hazards and Earth System Sciences
EGU
Discussions

410 $x$ Slope-parallel coordinate (downstream) [m]

$x_{\mathrm{CoM}}$ Position of the centre of mass (downstream) [m]

$y$ Slope-parallel coordinate (cross-stream) [m]

$z$ Slope-normal coordinate [m]

Natural Hazards and Earth System Sciences
EGU
Discussions

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

## Appendix A: Parameter values

Table A1 summarises the main variables considered for the parametric study presented in this paper. Table A2 gives the input parameters for the Perlin noise used in the generation of the complex topography.

| Variable | Values |
|---|---|
| Mesh resolution | 40 × 20 |
| | 160 × 80 |
| | 320 × 160 |
| | 640 × 320 |
| | 960 × 480 |
| Terrain roughness | simple / complex* |
| Cohesion ($\tau_c$) | 1, 10, 100 (Pa) |

**Table A1.** Variables considered for the parametric study. *Complex topography generated using Perlin noise.

| No. | Octave | Magnitude | Seed |
|---|---|---|---|
| 1 | 1 | 3 | 3 |
| 2 | 3 | 4 | 2 |
| 3 | 6 | 1.5 | 4 |
| 4 | 12 | 1 | 4 |
| 5 | 20 | 0.1 | 1 |

**Table A2.** Input parameters used to create the complex topography with the Perlin noise generator. Octave relates to the frequency band (small- versus large-scale structures); magnitude is the multiplier applier to each octave; and the seed could allow different realisations with identical statistical properties to be created.

## Appendix B: Low resolution results

This Appendix complements the high-resolution results shown in §2. Specifically, Figs. B1, B3 and B2, obtained with a mesh resolution of 40×20, can be compared with the higher-resolution cases (640×320) shown in Figs. 4, 5 and 6. Fig. B1 shows that low-resolution simulations lead to larger lateral and downstream spread of the flow, even for high values of $\tau_c$. Fig. B2 shows that reducing the resolution leads to much less material becoming static on the complex topography with low values $\tau$. Finally, Fig. B3 shows that the velocity distribution within flows tends to narrow for higher $\tau_c$ at lower resolutions, which is consistent with the high-resolution results. This underscores the interplay between the resolution, terrain complexity, flow cohesion and the *yielding* criterion.

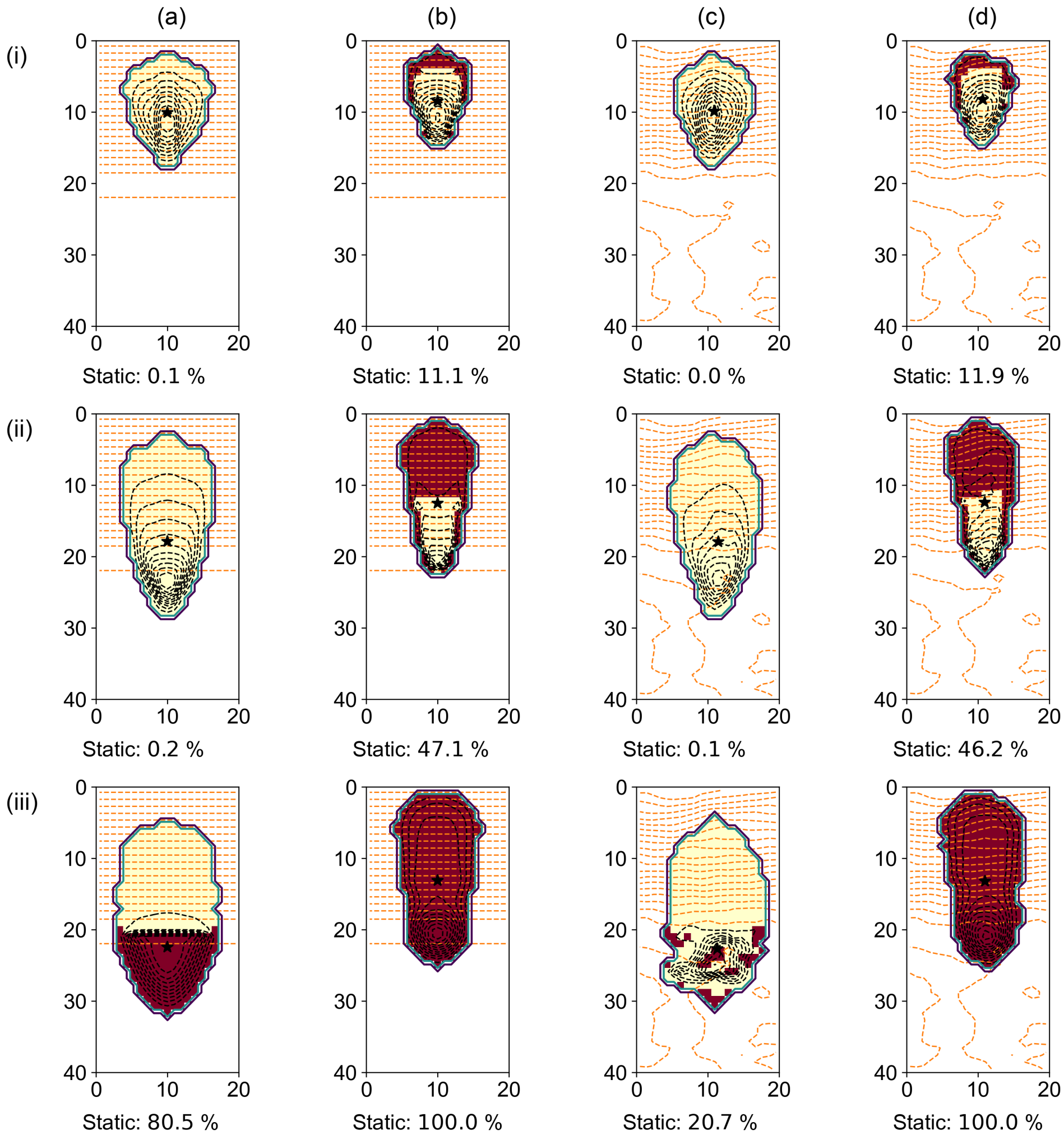

**Figure B1.** Flow snapshots for simple and complex topographies and a low mesh resolution of 40×20. (a) $\tau_c$ = 1 Pa; simple topography; (b) $\tau_c$ = 10 Pa; simple topography; (c) $\tau_c$ = 10 Pa; complex topography. (d) $\tau_c$ = 10 Pa; complex topography. The times for parts (*i*), (*ii*) and (*iii*) are $t = 2, 4$ and 8 s, respectively. Compare with Fig. 4.

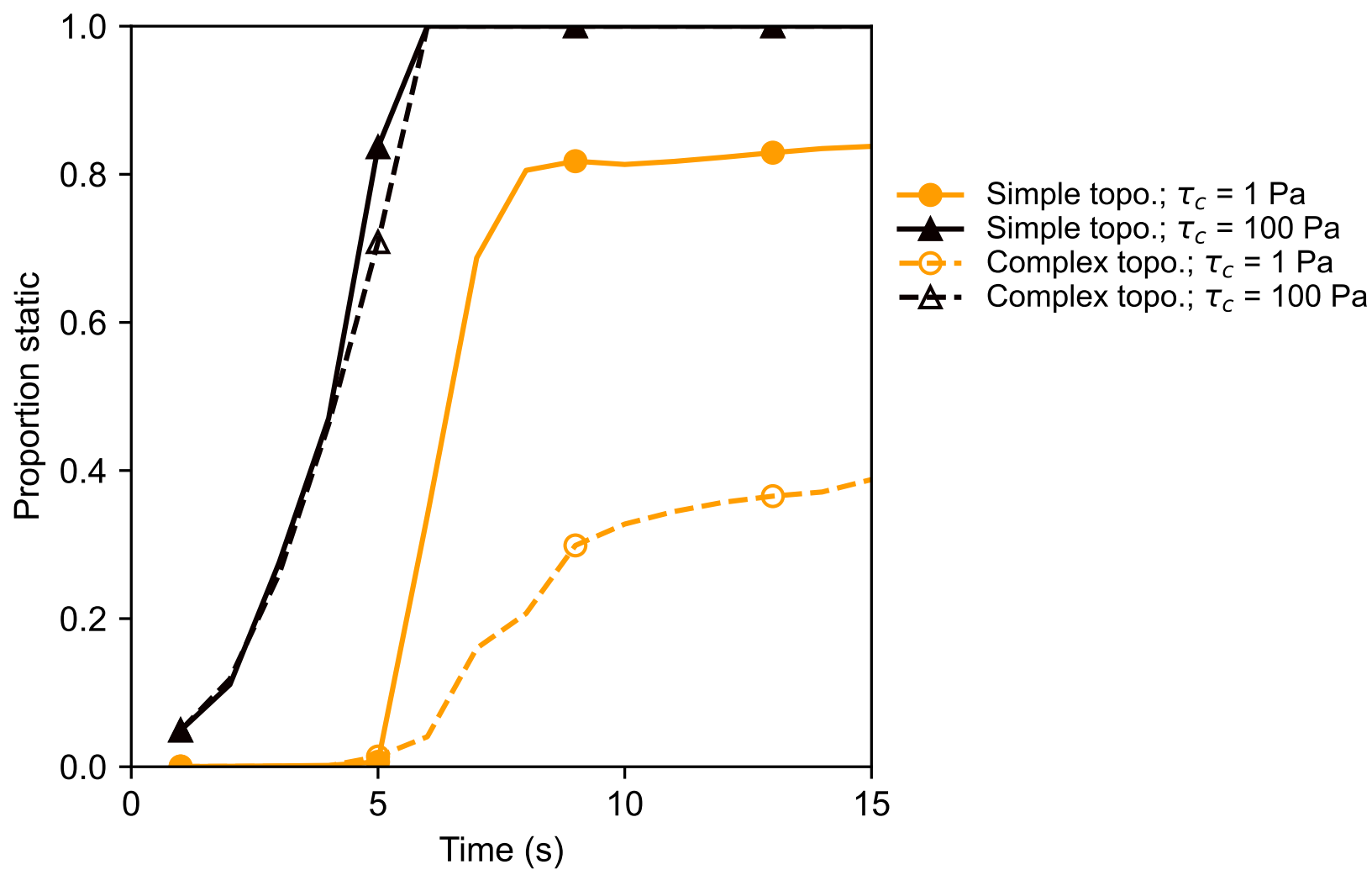

**Figure B2.** Proportion of static material as a function of time for a low mesh resolution of 40×20. Compare with Fig. 5.

## Appendix C: Internal stress distribution in flows

Fig. C1 shows a breakdown of the basal shear stress into the cohesion, Coulomb (dry) friction, and turbulent friction components. Two different cohesion values ($\tau_c = 1$ and 10 Pa) and the two topographical conditions (simple and complex) are shown. For the simple topography with $\tau_c = 1$, cohesion already contributes to more than 50% of the total stresses along the length of the flow at $t = 2$ s (Fig. C1a(*i*)). These cohesive stresses are minimal near the flow front, where the avalanche is fastest and deepest. This proportion subsequently increases, cohesion contributing to more that 80% of the stress for $t = 4$ and 8 s (Figs. C1a(*ii*) and C1a(*iii*)). Contributions of Coulomb friction and turbulent friction stresses progressively reduce in importance as the flow slows. Note that the Coulomb friction stresses are higher in the runout zone than on the slope. Turbulent friction contributes proportionately more to the stresses while the avalanche is in rapid motion, as expected. Similar results are obtained on the complex topography for $t = 2$ to 4 s (Figs. C1a(*i*) and C1a(*ii*)), even if the topography introduces some fluctuations into the profiles. At $t = 8$ s (Fig. C1a(*iii*)), the fluctuations are magnified, reflecting the specificities of the underlying terrain. For $\tau_c = 10$ Pa, cohesion contributes to more than 90% of the stresses along the flow length already at flow startup (Fig. C1b). In this case, the topographical complexity influences the flow dynamics less, due to thicker flows and reduced flow spreading.

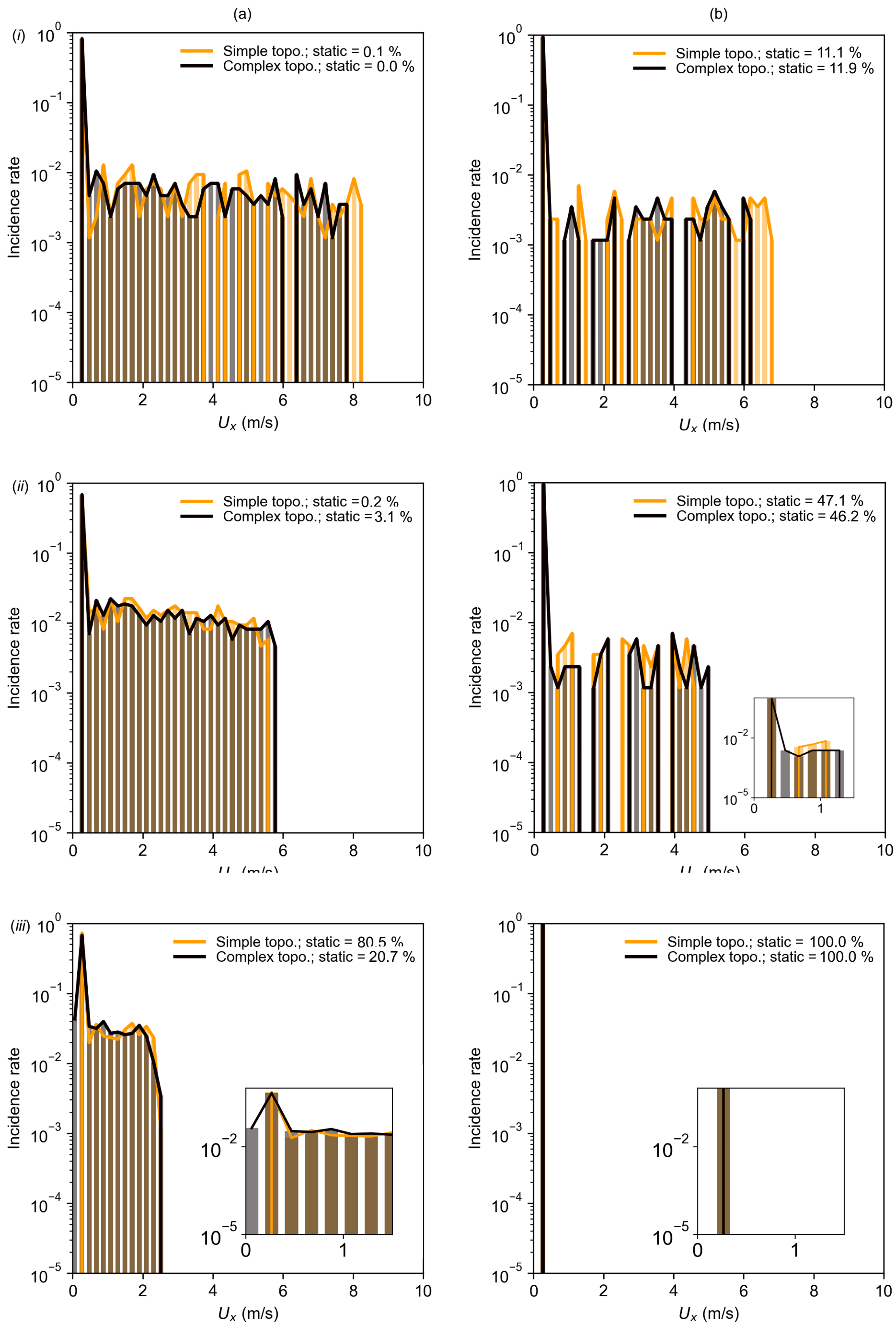

**Figure B3.** Velocity distributions for a low mesh resolution of x×xx. (a) $\tau_c$ = 1 Pa; (b) $\tau_c$ = 100 Pa. The times are (*i*) 2 s, (*ii*) 4 s and (*iii*) 8 s. Compare with Fig. 6.

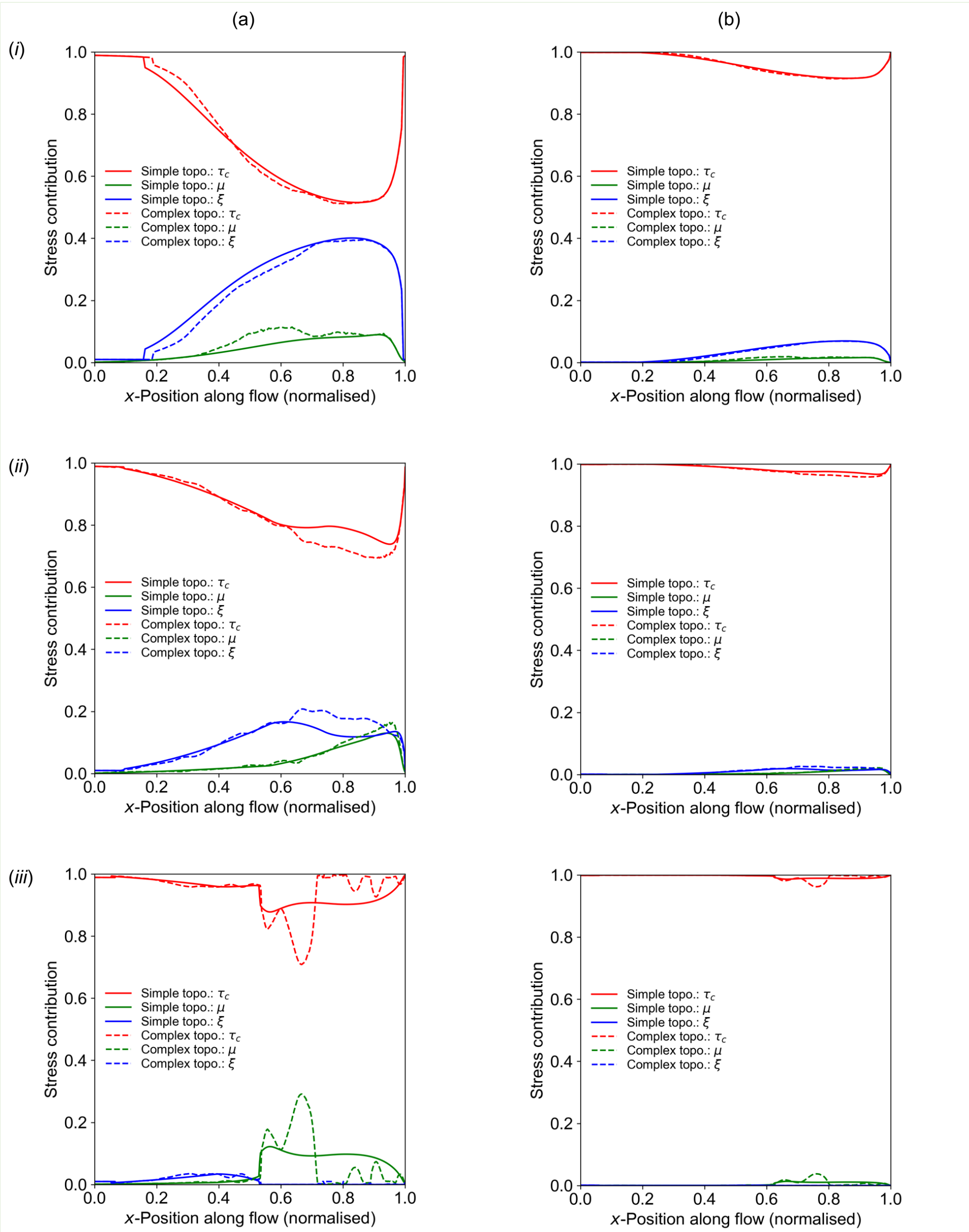

**Figure C1.** Breakdown of basal shear stress along a longitudinal cut of the flow and at different times. (a) $\tau_c$ = 1 Pa; (b) $\tau_c$ = 10 Pa. The times are (*i*) $t = 2$ s, (*ii*) $t = 4$ s and (*iii*) $t = 8$ s.

## Appendix D: Relationship between velocity and local inclination

Fig. D1 shows the same velocity distributions as Fig. 6, re-plotted as a function of the $x$-inclination of each node. At time $t = 2$ s (Figs. D1a(*i*) to D1c(*i*)), the horizontal lines observed for the case of simple topography reflects the uniform slope. For the complex topography, similar velocity/angle distributions are observed for different cohesion values $\tau_c$, due to the prevalence of inertial effects at this early stage of the flow. At $t = 8$ s, for $\tau_c = 1$ Pa (Fig. D1a(*iii*)) and the simple topography, the relationship between velocity and slope is well defined, and no points where $\theta < 7°$ are moving; this is close to the basal friction angle of $9°$, and it is expected that the cutoff angle should continue to approach the basal friction angle as more time elapses. For the corresponding complex topography, the relationship between velocity and slope varies more. Notably, even where $\theta < 9°$, $U > 1$ m/s for many points. Fig. D1b(*iii*) is similar, although $12°$ is the cutoff for the simple topography, due to higher $\tau_c$. In Fig. D1c(*iii*), $U \approx 0$ m/s for all points, consistent with Fig. 6c. These graphs highlight the combined effects of $\mu$ and $\tau_c$ on the flow dynamics, relative to the underlying topographical inclination.

Natural Hazards and Earth System Sciences
EGU
Discussions

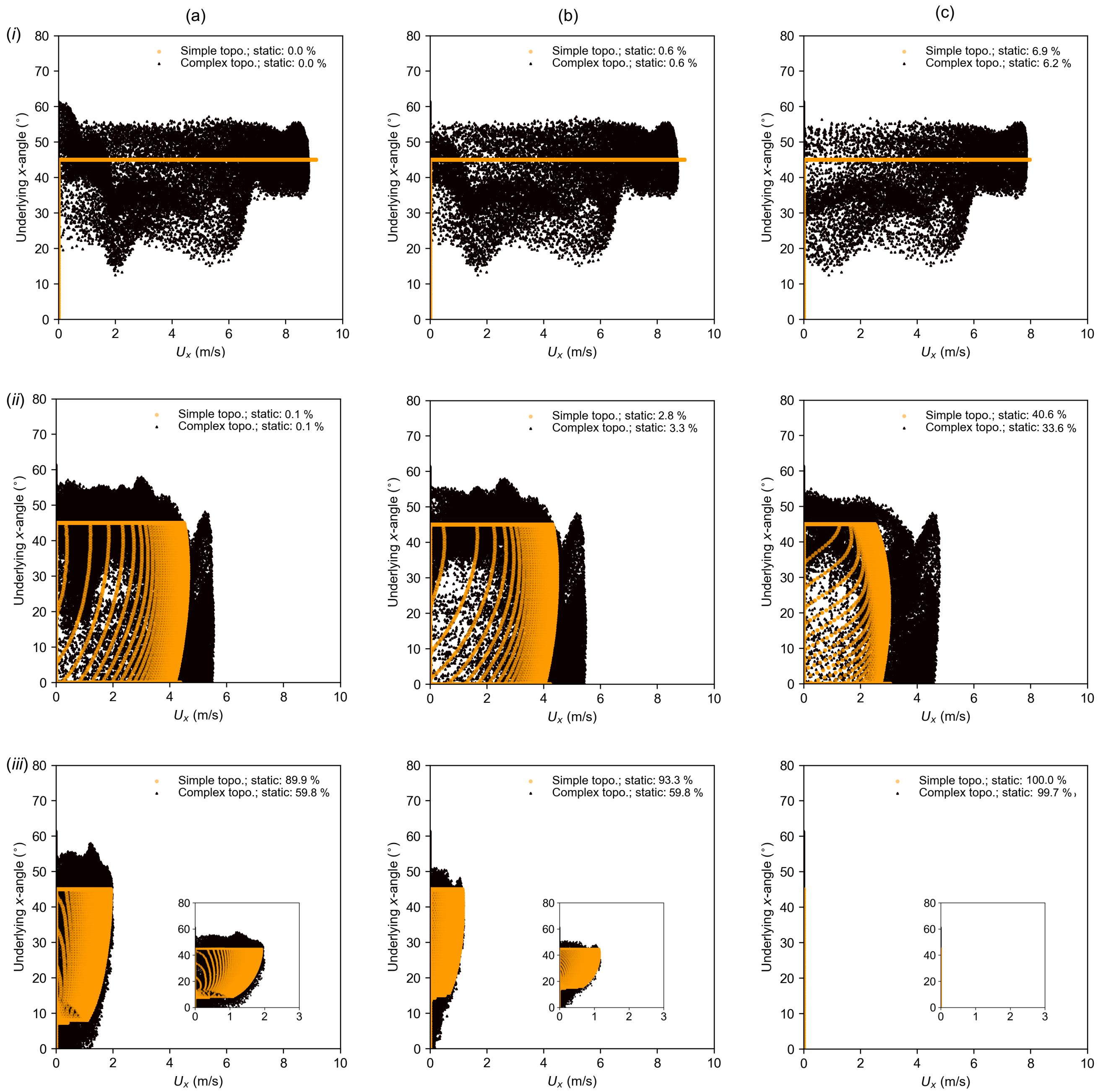

**Figure D1.** Scatter plot of local $x$-velocities versus the underlying $x$-angles of the topography. (a) $\tau_c$ = 1 Pa; (b) $\tau_c$ = 10 Pa; and (c) $\tau_c$ = 100 Pa. The times are (*i*) $t = 2$s, (*ii*) $t = 4$s and (*i*) $t = 8$s. Data for both simple and complex topographies are represented in yellow and black respectively. The insets show zoomed-in versions of the same data.