# Peer review of "Has it *really* stopped? Interplay between rheology, topography and mesh resolution in numerical modelling of snow avalanches"

_Natural Hazards and Earth System Sciences, 2024_

## Author Comment (AC1)

**1    Reviewer #1**

**Q1** — The manuscript describes the extension of a depth-averaged flow model that uses Voellmy's rheology by a physically-based stopping criterion. To be honest, I am not convinced by the manuscript in its present form. There are too many aspects not explained or justified sufficiently.

5      From my point of view, publishing the manuscript as a research paper would require much more explanation of the approach and convincing arguments that the problem addressed here is important for practical applications and that it is a general problem and not just a deficiency of the implementation used here.

Since these points are quite fundamental for me, I do not write line-by-line comments at the moment, but look forward to reviewing a revised version. I would also like to point out that my rating of the manuscript refers to the "worst-case scenario"
10   and would be much higher if the points raised above can be addressed.

**A1** — We appreciate you having taken the time to review our manuscript. Please refer to our point-by-point responses below.

**Q2** — There is only a vague promise to disclose the codes used for the simulation and for the analysis after acceptance of the
15   paper. This makes assessing the correctness of the manuscript difficult.

**A2** — Thank you for your helpful suggestion. We shall add as an Appendix a full description of the code.

**Q3** — It does not become clear whether the issue addressed in the paper is a general problem of Voellmy's rheology (and then
20   also of similar rheologies) of just a deficiency of the implementation used here (and perhaps of some other implementations). In the beginning of Sect. 2.2, it reads as if Voellmy's rheology itself would not let the material come to rest. This is not true since material will finally come to rest if the tangent of the slope angle of the free surface is smaller than the coefficient of friction $\mu$. This means that spreading of the deposits in the runout zone should stop completely. Otherwise, there may be a problem with the numerical implementation of the friction term. I was involved in two model developments. One of them (based on Gerris,
25   doi 10.5194/nhess-15-671-2015) lets the fluid accelerate first an the uses a fully implicit scheme for the friction term, which leads to a permanent creeping and some artificial spreading of the deposits. The other (MinVoellmy, doi 10.5194/gmd-17-781-2024, already cited) uses a mixed scheme for the friction term and stops reasonably well in the runout zone. I do not know whether the model used here includes a 'good' or a 'bad' implementation of the friction term.

30   **A3** — Yes, we agree that the material will "finally come to rest" using a Voellmy rheology, when the material reaches sufficiently gentle gradients and provided a proper yielding criterion is implemented. One issue is *the time involved for this to occur* – it is not computationally tractable to wait for all nodes in the model to converge to zero when performing e.g. sensitivity studies. Furthermore, if you let the model run for too long the results become unrealistic due to numerical diffusion, it is important to be able to identify the point at which the flow can be considered as arrested, to discard all results that would

be obtained later. Our intention was to start from the point of view that practitioners want to optimise their model run time, rather than increasing it by some arbitrary factor to try to wait for convergence to zero. We note that the original text from the Introduction was not sufficiently clear on this subject:

> Numerical diffusion affects these numerical models, the extent depending typically on the numerical scheme adopted, the friction law used, and the topographical complexity (see also Hergarten, 2024). In practice, this numerical diffusion can lead to excessive spreading of avalanche deposits, smoothing of fronts, and, relatedly, persistent non-convergence of velocities to zero after apparent macroscopic flow stoppage. These artifacts are generally caused by numerical discretisation of the flow domain and numerical handling of floating point numbers.

We would therefore update this text to discuss the run-time issue. A secondary problem is the influence of numerical diffusion due to complex topographies, which we believe to affect all schemes on complex topographies, and which induces non-physical contributions to the results. We propose making changes to the wording of Section 2.2 as well, to further clarify that "not coming to rest" is true for "complex topographies" and "under tractable computational times".

**Q4** — It is true that some models include an empirical (non-physical) stopping criterion. However, there is no illustration or discussion how strong the effect of such a criterion is practically and whether it might be relevant for hazard assessment.

**A4** — As mentioned in the current version of our manuscript:

> RAMMS2D, for instance, implements several options for user-defined thresholds to terminate simulations, the most commonly used considering the ratio between current global flow momentum and the maximum global momentum reached during the flow. Zugliani *et al.* (2021) suggest using typical values between 1 and 10 % for this threshold.

Although momentum and velocity are not directly interchangeable, some general insights can still be obtained by considering the above thresholds (for momentum) in the context of velocity. For instance, for a flow with a maximum velocity at the upper bound recorded for avalanches of 100 m/s, a momentum threshold of 10% would imply an average residual velocity on the order of around 10 m/s. Depending on the underlying topography, this could translate into an additional runout distance of tens (or hundreds) of metres, beyond the final state of the simulated avalanche. Even with a momentum threshold of 1%, this would give an average residual velocity on the order of 1 m/s. This could also lead to substantially longer runout distances than what is computed, which might be significant in terms of risk assessments, or influencing where one might want to install mitigation measures. We propose adding text into the manuscript to discuss this in more detail.

**Q5** — The stopping criterion (Eq. 6) is not explained completely since a proper definition of $\tau_{b,\text{test}}$ is missing. Is the criterion just that the actual momentum could be consumed entirely in the actual time step, as implemented in MinVoellmy? Or is it something more elaborate?

**A5** — We agree that this was a deficiency of the original manuscript. We shall define $\tau_{b,\text{test}}$ mathematically in the Appendix. The idea behind the criterion is that if the resisting stresses (encapsulated in $\tau_c + \rho g_z h \mu$) are greater than the driving stresses (encapsulated in $\tau_{b,\text{test}}$), the flow velocity is set to exactly (integer) zero instead. Without this criterion, a flow velocity of near-zero would be achieved instead, with a stronger asymptotic component than with the criterion.

**Q6** — The authors often refer to numerical diffusion as the source of the issues addressed in this manuscript. This is not clear to me, in particular since the simple implementation in MinVoellmy does not include any measures against numerical diffusion, but shows little spreading of the deposits.

**A6** — Thank you for pointing out the lack of sufficient clarity here. As mentioned in Hergarten (2024), numerical diffusion is a key issue for CFD solvers, although this paper also states that for the MinVoellmy implementation, numerical diffusion is not a major issue. However, as far as we can tell, the topographies considered in that manuscript were simple, i.e. were comprised of planes and splines. We do not debate that for simulations on *simple* topographies, most implementations show little spreading of the deposits – which is consistent with the results from our current scheme. The intent of the manuscript is to highlight how spreading continues indefinitely on *complex topographies*, which is an issue that affects multiple numerical schemes into which the Voellmy rheology is implemented. We note that some of the text from the Introduction could probably be refined. For instance:

> Numerical diffusion affects these numerical models, the extent depending typically on the numerical scheme adopted, the friction law used, and the topographical complexity (see also Hergarten, 2024). *In practice*, this numerical diffusion can lead to excessive spreading of avalanche deposits, smoothing of fronts, and, relatedly, persistent non-convergence of velocities to zero after apparent macroscopic flow stoppage.

We would update "In practice" to explictly refer to complex topographies, and also clarify the "see also Hergarten, 2024" to give more relevant details, per the above discussion.

We also note that numerical diffusion is evaluated within Figures 9 and 10, and their associated text, in the original version of the manuscript. We propose adding more cross-references throughout the manuscript to help guide the reader.

**Q7** — As a second aspect, the manuscript discusses the effect of an additional cohesion term in Voellmy's rheology. In contrast to the points discussed above, this aspect becomes clear, but is not very surprising. As mentioned above, Voellmy's rheology without cohesion lets the material move permanently if the fluid surface is steeper than the tangent of the coefficient of friction $\mu$. This results in an ongoing, but small flow into the runout zone and finally lets the runout zone grow a bit. Cohesion just results in an increase of $\mu$ by a term $\tau_c/(\rho g_z h)$, which is inversely proportional to the thickness $h$. So flow on the slope stops if the thickness falls below a threshold that depends on cohesion.

**A7** — Yes, this is an accurate summary of this part of the paper. However, the ability of the cohesion term to allow us to isolate the effects of numerical diffusion was valuable. Furthermore, it shows that our numerical scheme is able to properly cause the flow material to converge to zero velocity, even for flows on complex topographies. This implies that flow material with a lower cohesion would also converge to zero velocity, but over a much longer period of time. We would adjust the discussion within the manuscript to put further emphasis of the value of the results relating to cohesion being related to the isolation of numerical diffusion on complex topographies. Furthermore, for practical applications, cohesion "helps" to define flow arrest even in presence of numerical diffusion

---

## Author Comment (AC2)

**2 Reviewer #2**

**Q1** — In this paper, the authors aim to define criteria for when material truly stops in depth-averaged models, distinguish between physical stopping mechanisms and issues related to numerical diffusion, and discuss or propose best practice guidelines to address these challenges. The manuscript examines the interplay between three key variables affecting avalanche arrest: mesh resolution, topographic complexity, and snow cohesion. The authors use a second-order depth-averaged model as a test framework, incorporating a modified Voellmy model with cohesion and a physical yielding criterion. To the best of the reviewers' knowledge, this complete form of the model has not been tested before.

Defining criteria for when material truly stops in depth-averaged models is challenging from both physical and numerical perspectives. Numerical diffusion depends on factors such as the numerical scheme, grid resolution, and time-stepping methods. Additionally, physical stopping mechanisms are a matter of debate and vary from model to model. The choice of stopping condition is closely linked to the specific output the model is intended to produce. As a result, there isn't a one-size-fits-all criterion for determining when material has truly stopped moving in a numerical model. Each model's unique characteristics mean that numerical diffusion can have varying impacts, requiring tailored criteria to accurately assess stopping conditions. This issue also extends to different mechanical deposition criteria.

In this respect, I'm not yet convinced that the paper will have a significant impact on the international community, as the results appear specific to the models used by the authors and lack broader evidence of global relevance. The proposed global arrest criteria are not entirely new or convincing. For example, the role of calculation grid resolution has already been established in previous work, and the other criteria can also be questioned (see comments below). The authors need to strengthen their arguments in this regard.

Furthermore, I feel the paper lacks a discussion or analysis regarding the overall relevance of the stopping issue in relation to the final outputs. Ultimately, what really matters is how much the runout distance or the impact pressure at a specific location can be influenced by these issues. Understanding this impact is crucial for assessing the practical implications of the stopping criteria.

Finally, although the paper's main aim is not focused on the model's performance, the extensive description of the model and the impact of cohesion raises questions about the physical appropriateness of the model. For example, the cohesion values used in the simulations (1-100 Pa) are relatively small compared to the broader range of possible values mentioned in the paper (0-2300 Pa). Despite this, the model shows a strong response to variations in cohesion. Is this behavior physically realistic? This is particularly important to discuss because cohesion take an important role in your conclusions.

**A1** — Thank you for spending the time to review our manuscript, and for providing a set of detailed comments. To address some of the issues mentioned above:

- We believe that the text in our original manuscript relating to the global arrest criteria could be productively ameliorated throughout – it is not really our intention to suggest that any such criteria are a good way of defining simulation stoppage. In contrast, we believe the locally-based criteria to be a novel, and potential robust, stepping stone towards more

rigourously defining when simulations have finished. We propose clarifying the manuscript throughout to highlight this. Furthermore, the manuscript is intended as a "proof of concept" for taking an approach to evaluate the effects of numerical diffusion under a matrix of variables including the grid resolution, the topographical complexity and the cohesion. We state that this methodology should be adopted to perform a sensitivity study for each model, considering the likely range of parameters encountered, in the current version of the text:

> Finally, let us recall that this study only considered simple avalanches on model topographies. Similar sensitivity analyses on real full-scale topographies will be needed to further ascertain the conclusions and, in particular, validate the applicability of the proposed local arrest criterion. In real cases, processes such as splitting of the flow into several branches, or impacts against counter-slopes, may further complicate the detection of avalanche arrest. Along a similar line, the influence of the dry and turbulent friction coefficient, $\mu$ and $\xi$, which were here maintained constant, would need to be investigated in conjunction with variations in cohesion $\tau_c$.

We thus propose updating the text of the manuscript to emphasise that we have novelly developed a proof-of-concept for assessing the sensitivity of numerical schemes across a matrix of relevant parameters (in addition to having novelly developed a local stopping criterion which may be more objective than the globally-based criteria available to date).

– We agree that the original version of the manuscript is not sufficiently self-contained, in that we do not provide specific examples of how much the runout distance may be affected at full scales. We propose updating the Introduction to discuss one or two papers from the open literature, to expand on what is mentioned in the current version of our manuscript:

> RAMMS2D, for instance, implements several options for user-defined thresholds to terminate simulations, the most commonly used considering the ratio between current global flow momentum and the maximum global momentum reached during the flow. Zugliani *et al.* (2021) suggest using typical values between 1 and 10% for this threshold.

Although momentum and velocity are not directly interchangeable, some general insights can still be obtained by considering the above thresholds (for momentum) in the context of velocity. For instance, for a flow with a maximum velocity at the upper bound recorded for avalanches of 100 m/s, a momentum threshold of 10% would imply an average residual velocity on the order of 10 m/s. Depending on the underlying topography, this could translate into an additional runout distance of tens (or hundreds) of metres, beyond the final state of the simulated avalanche. Even with a momentum threshold of 1%, this would imply an average residual velocity on the order of 1 m/s. This could also lead to substantially longer runout distances than what is computed, which might be significant in terms of this part of the risk assessment, or influencing where one might want to install mitigation measures. We propose adding text into the manuscript to discuss this in more detail.

– We also commit to discuss the scaled-down cohesion in the paper more clearly. We note that in the original version of the manuscript, we discuss values of cohesion firstly in lines 84 to 87, and then again in lines 128 to 134, which is

confusing. We would direct all discussion of the cohesion towards the text already in the manuscript relating to non-dimensionalisation of the cohesion:

We also define the dimensionless ratio between cohesive and gravitational stresses:

$$N_c = \frac{\tau_c}{\rho g_z h}. \tag{1}$$

The maximum value reached in our simulations was $N_c \approx 0.15$, for typical heights $h \approx 0.2$ m in the runout zone. Note that such values of $N_c$ would correspond to typical cohesion values in the range 0.5–2.5 kPa at full scale (for $1 < h < 5$ m), consistent with recent studies on wet snow... The number $N_c$ exemplifies the inherent scale-dependence of the cohesion parameter $\tau_c$.

**Q2** — Abstract, line 1: I disagree with the assertion that "depth-averaged models of snow avalanches have hitherto lacked an objective arrest criterion." The paper does not adequately test or consider existing methods—many of which are mentioned in your bibliography but not further analyzed—making such a blanket statement misleading.

**A2** — Thank you for your comment. We accept that whilst we evaluate some methods used for determining when simulations may be stopped, that we are not comprehensive. We do not, for instance, test stopping criteria based on momentum, which are also present in RAMMS. For stopping criteria for momentum specifically, we do not believe that they would give fundamentally different results from those based on velocity, due to the interplay betwixt complex topographies and numerical diffusion. We propose amending the original assertion in the abstract to state that "Developing objective arrest criteria for depth-averaged models of snow avalanches is challenging".

**Q3** — Furthermore, it's important to clarify what differentiates an objective criterion from a subjective one. For instance, a limit based on the physical threshold of a variable is, in my opinion, also an objective criterion.

**A3** — We shall amend the text to clarify that "objective" is intended to mean "(1) based on a specific physical threshold (e.g. velocity being set to zero) and (2) does not affect simulation results through tweaking arbitrary parameters.

**Q4** — Figure 2: The captions are generally very small; please ensure they are easily readable. Additionally, the legend describing the green-brown color is missing.

**A4** — We shall make consistent all the figures, and will furthermore embiggen the caption size.

**Q5** — Line 136: At the beginning of a sentence, 'Figs. 3a' should be written out in full as 'Figure 3a' for proper academic style. This issue occurs multiple times throughout the paper, so I suggest a thorough review.

210 **A5** — We shall change all instances of "Fig." to "Figure".

**Q6** — Line 137: How are the center of mass and the tip of flow defined (CoM) exactly? In depth-averaged modeling, defining the avalanche tip and the center of mass is not entirely straightforward and can be complex due to several factors.

215 **A6** — We agree that these points were not properly elucidated in the original version of the manuscript, and undertake to clarify them. The centre of mass of the flow is computed using the flow mass distributed at each node (whose spatial coordinates are known) within the system. Specifically, we sum the product of each node of mass and its distance from the (0,0) coordinate, and then normalise by the total weight of material. The frontal tip of the avalanche is defined as the further downstream node (in the $x$-direction) which has non-zero material associated with it.

220

**Q7** — Line 138: you mean substantial flow stoppage? not substantive

**A7** — Yes, we shall change this. Thank you.

225 **Q8** — Figure 3. The figure shows 6 visible lines, but there are 10 items in the legend, for each graph. Please mention in the caption that some of the curves are overlaid? Or are they just missing?

**A8** — Yes, some of them are overlaid, as they have converged. We shall update the caption accordingly.

230 **Q9** — Line 139: It is observed that grid sizes of 320×160 and coarser significantly influence simulation results. Do you mean 160x80? I think the 320x160 is not visible in the graphics. Additionally, the 160x80 grid shows only around a 2% difference, which is relatively minor. Providing these percentages helps clarify the magnitude of the error we are discussing.

**A9** — Thank you for pointing out this error – you are correct, it should read "$160 \times 80$ and coarser". We would also adjust the
235 wording of "significantly influence" to "influence" and note the percentage difference between the cases. We also commit to adding a comment in the figure caption stating which datasets are overlaid, and are hence not visible, per the previous response.

**Q10** — Lines 147-150: The flow is described as stopping "from the bottom up." Could you clarify whether this refers to a back-propagating shock or if you are indicating that material stops first on gentler slopes and, in more cohesive scenarios, even
240 on steeper slopes? The terms "bottom-up" and "bottom-down" seem unclear and could be misleading in this context.

**A10** — We apologise for the lack of clarity in the manuscript. We are not referring to back-propagating shocks, although such back-propagating shocks are indeed observed in some cases, when cohesion is (very) low!. Our intention was to indicate that

material stops first on gentler slopes, unless the cohesion is so high that it can start depositing immediately, even on steeper slopes. We propose revising the text to properly clarify the deposition mechanisms for each set of cohesions.

**Q11** — Caption figure 4: (f) $\tau_c = 10$ Pa should be $\tau_c = 100$ Pa. I'm wondering: The cohesion values used in your simulations (1–100 Pa) are relatively small compared to the broader range of possible values mentioned in the paper (0–2300 Pa). Despite this, the model shows a strong response to variations in cohesion. Is this behavior physically realistic? Could you provide an explanation for why the model reacts so significantly to these relatively small changes in cohesion?

**A11** — Firstly, thank you for pointing out the typo. Secondly, the model is at a reduced scale, so relatively small changes in cohesion can indeed cause significant changes in results. The effect that a given change in cohesion will cause is encapsulated in the dimensionless ratio $\tau_c/(\rho g_z h)$; see also the discussion in **A1**. We do not wish to expand the scope of the manuscript further to consider full-scale cases, and this is left for future investigations.

**Q12** — Line 203: As shown in the previous section, relying on the physically-based yielding criterion is generally insufficient for defining avalanche arrest objectively when a realistic, complex topography is considered. These assertion may not be entirely accurate. This conclusion could be influenced by the specific yielding criterion used in this study, which might not be the most suitable for the scenarios considered. Consequently, this raises questions about the extent to which the results obtained from this analysis can be generalized to other models. This need to be discussed in the paper.

**A12** — We emphasise that only the exercise performed in this paper is generalisable to other schemes, i.e. one has to re-peat the sensitivity study for each numerical scheme, to assess how much numerical diffusion influences results on complex topographies. Some discussion on this is already provided at the end of the main text in the original version of the manuscript:

> Finally, let us recall that this study only considered simple avalanches on model topographies. Similar sensitivity analyses on real full-scale topographies will be needed to further ascertain the conclusions and, in particular, validate the applicability of the proposed local arrest criterion. In real cases, processes such as splitting of the flow into several branches, or impacts against counter-slopes, may further complicate the detection of avalanche arrest. Along a similar line, the influence of the dry and turbulent friction coefficient, $\mu$ and $\xi$, which were here maintained constant, would need to be investigated in conjunction with variations in cohesion $\tau_c$.

Benchmarking multiple numerical schemes is outside the scope of this paper, but could be a valuable future contribution; this would also help answer the additional scientific question that you raise, i.e. "what yielding criterion is most appropriate for which case?"

**Q13** — Figure 5 and Lines 168-178: In my understanding the higher the cohesion, the slower should be the flow, the longer should be the simulations time to allow all material to come to a natural rest. This effect is reinforced on complex topography

because the random acceleration and decelerations phases. Therefore, it is crucial to define when the avalanche has truly physically stopped, before compare results. For example: What happens to the curves with complex topography and high cohesion if the simulation is extended to 30 seconds? Specifically, how much time does it take for all the material to reach the flat slope and come to a complete stop? At what point does numerical dissipation begin to affect the results? I do not believe this distinction is adequately addressed in the paper.

**A13** — This is not quite correct – the higher the cohesion, the slower the flow, and so *shorter* simulation times can be adopted to allow all material to come to a natural rest.

As for the other part of this comment: if simulations with high cohesion and complex topographies were further extended, numerical diffusion would lead to a continued asymptotic decline in the local flow height, as implied by Figure 10. However, the purpose of the manuscript is not to determine the length of time required for simulated flows to come to "full" stoppage under various combinations of conditions. We note that the results from the high cohesion flows strongly suggest that it is possible for our numerical scheme to be able to capture "full" stoppage, even on complex topographies, under short timescales. We therefore believe it to be a reasonable extrapolation to assume that the same is true for low cohesion flows, given a sufficient simulation time. However, determining what constitutes "sufficient time" is not helpful for design practice, where limited computational resources must be spent to evaluate multiple hypothetical scenarios. Furthermore, running for arbitrary periods of time would not, on its own, distinguish the effects of numerical diffusion from physically-based flow movement; and if you let the simulation run for too long, the final outcome (runout, height). will become really unrealistic due to diffusion. Finally, our model is run at a reduced scale, and developing scaling relationships for the variables considered in this manuscript, in terms of their effect on the computational time, would significantly expand the scope of this work.

All of this motivated us to design our manuscript to focus instead on a simple method of defining "early on" when the simulation can be deemed as terminated. Our proposed criterion is given in the text relating to Figures 9 and 10. There, the state of representative points in the flow becoming "unyielded", whilst the corresponding point in the deposit continues to change in depth, is indicative of numerical diffusion becoming the dominant contributor to further changes in the flow. It can only be assumed that the numerical diffusion, before the deepest point of the deposit becomes unyielded, contributes minimally to the flow. Note that an indication of the relative magnitude given by comparing the rates of the change of the depth of the deposition before and after the representative point transitions from being "yielded" to "unyielded".

We propose incorporating some of this discussion into a revised version of the manuscript, regarding such effects of numerical dissipation at different stages of the simulation.

**Q14** — Line 183 At time $t = 2$ s, is it observed that cohesion $\tau_c$ and the topographical complexity only weakly affect the results... The maximum difference is around 1 m/s ( 8 to 9 m/s velocity maximum) corresponding to around 10%., so, not so small.

**A14** — Thank you for pointing this out. We shall clarify the text here to state that the effect is not weak.

**Q15** — Line 225-226. "Potentially, it [using averaged velocities to define arrest] could thus offer a more objective criterion to define effective flow arrest, although it would still require to be manually pinpointed on the curves." This would mean applying a different criterion for each individual simulation and scenario, which may not simplify the process. I do not believe this would make things easier or that it represents an applicable rule.

**A15** — We agree with your statement, and will add text to clarify that we do not believe this to fulfill our newly-defined concept of objectivity.

**Q16** — Line 251-252. For simple and complex topographies, Figs. 9 and 10 show time-histories of different metrics relating to the highest point of the flow material on the runout zone: It is unclear what is meant by "the highest point of the flowing material in the runout zone". Are you referring to the location of the point with maximum flow or deposition depth at each time step in the lower part of the track ($x > 20$ m)?

**A16** — We apologise for the lack of clarity in the original version of the manuscript, and note that the confusion so caused also gave rise to some of your other comments. In a revised version of the manuscript, we would define it more clearly: it is "the highest point of the material in the runout zone ($x > 20$ m) at any given timestep, such that this point is not necessarily at the same location for all datapoints plotted, even for a single modelled avalanche event".

**Q17** — Figure 9 shows metrics related to the highest point (hp). However, the figure does not specifically reference the 'highest point' but rather shows flow depth. It would be clearer to use the term "maximum flow depth" in the caption to accurately match what is presented in the figure. This change should also be reflected in the main text.

**A17** — We agree with your comment and shall update the figures and text accordingly.

**Q18** — Additionally, it is unclear why mesh resolution is considered again, given that its major impact was already established. Furthermore, the symbols for the curves, especially for velocity, are difficult to distinguish. Improving the differentiation of these symbols would be helpful. Only two subplots are labeled with letters. All subplots that are discussed should be labeled, and each description should clearly indicate which subplot it refers to.

**A18** — We agree that the motivation for re-introducing the mesh resolution at this part of the manuscript was not properly clarified in the original version of the manuscript. The idea was that we had only evaluated mesh resolution for "global" arrest criteria; we wanted to look again at these issues in the context of our proposed "local" arrest criteria, to establish whether "local" criteria might enable the adoption of a lower resolution than we had previously concluded. We also wanted to ascertain whether convergence for different resolutions inferred through "global" criteria would also be found through "local" criteria.

We propose inserting one or two sentences at the start of this section to introduce the motivation for re-considering the grid resolution.

350     We also commit to lettering the individual subplots, so as to help clarify which part is being referred to in the relevant parts of the text.

**Q19** — Lines 259-260: Why is the discussion returning to mesh resolution, which was already addressed and fixed in Section 3.1? The paper is complex enough, and mixing processes can lead to more confusion. It would be better to keep the discussions 355  on different processes as separate as possible.

**A19** — Please see the previous comment.

**Q20** — Lines 272-273: The statement suggests that certain metrics related to the highest point in the runout zone—specifically, 360  the first transition to a static state and the cancellation of its velocity—serve as objective criteria for defining avalanche arrest. However, I find this problematic. In real-world avalanches, the first point that stops in the runout zone can be overrun by material still in motion along the avalanche path, often reaching the deposition zone with a slight delay compared to the leading front. Therefore, this stopping criterion is only valid when the entire mass moves as a cohesive front, which is rarely the case in practice.

365

**A20** — Thank you for your valuable insights. We believe that our technique is not sufficiently clearly described in the original version of the manuscript. We monitor the depth of the deepest part of the avalanche material in the deposition area; this point can change with time, if further oncoming avalanche material interferes with the deposit. As such, this stopping criterion does not depend on the entire mass moving as a cohesive front. We propose adding your comments into the discussion of this 370  stopping criterion, to further clarify matters for the readership.

**Q21** — In depth-averaged modeling, "overrunning" cannot be directly accounted for, but material from behind can still push the already deposited material. Could this be a possible reason for the observed shift in maximum depth, as shown in Figures 9 and 10? If I'm misunderstanding, it might be due to the lack of straightforward visualization in this part of the paper. Perhaps 375  the authors could provide a video to better illustrate the processes involved?

**A21** — Yes, it is possible that material from behind can also contribute to changes in the depth of deposited material, although we would expect such effects to *increase* the depth of the deepest part of the deposit. (Recall that for these two Figures, we monitor the depth of the deepest part of the deposit.) We propose elaborating more clearly on this aspect in the paper. We also 380  agree that providing a video of the flows would also help the readership significantly, and would provided this with a revised manuscript.

13

**Q22** — Lines 278-279: The statement "However, sufficiently robust and objective arrest criteria are hitherto lacking" in the paper is questionable. Many models already have established stopping criteria, and the paper does not provide sufficient evidence to demonstrate their lack of robustness. Additionally, it is debatable whether truly universal robust criteria can be developed that are effective across all models, given the diverse nature of numerical diffusion and stopping conditions.

**A22** — We agree that it may not be possible to develop "truly" robust criteria that can apply across all numerical schemes and for all rheologies. The purpose of the present manuscript is simply to try to make improvements, and we can make further clarifications to delimit our contributions. As for the other part of your comment: Reviewer #1 has also requested further evidence that demonstrates the lack of robustness of existing arrest criteria, and we believe this to be demonstrable by referencing other papers. As mentioned in the current version of our manuscript:

> RAMMS2D, for instance, implements several options for user-defined thresholds to terminate simulations, the most commonly used considering the ratio between current global flow momentum and the maximum global momentum reached during the flow. Zugliani *et al.* (2021) suggest using typical values between 1 and 10% for this threshold.

Please see also the discussion in **A1**.

**Q23** — Lines 337-338: The claim that "mesh resolution affects simulated flow properties if the cell size is larger than about 20% of the characteristic flow depth, typically for both simple and complex topographies" cannot be generalized. These values are specific to your particular setup and should be clearly stated as such. Furthermore, this aspect is already well established in the literature.

**A23** — We agree that this comment should be properly delimited to our own study, and would need to be further evaluated for other simulated conditions, and shall update the text accordingly. We would also further clarify that this is a general value, even for our own study, since the effects of cohesion and terrain complexity also have some effects on the minimum mesh resolution possible, and would note also the limitation that we do not investigate this threshold for the mesh resolution very precisely.

We would furthermore remove the text corresponding to "20 %" from the abstract, as this is not a key contribution from our study.

**Q24** — Lines 341-345: The statement regarding large cohesion values promoting a full transition of the material to a static state and following lines, seems more aligned with the conclusions of a different study. The influence of cohesion on snow mobility is a distinct topic from the primary focus of your paper. Furthermore, numerical dissipation can be seen as a weaker effect compared to strong physical processes like cohesion. It is obvious that when cohesion is high, it plays a dominant role in stopping the material, reducing the significance of numerical dissipation. Dissipation effects may only become prominent when physical forces, such as cohesion, are weaker. Maybe you can discuss on that.

**A24** — Thank your for your helpful comment; this issue was also probed at by the other Reviewer, and is well worth addressing. The ability of the cohesion term to significantly shows that our numerical scheme is able to properly cause the flow material to converge to zero velocity, even for flows on complex topographies, using our numerical scheme. This implies that flow material with a lower cohesion would also converge to zero velocity, but over a much longer period of time. We would make further clarifications regarding the relative prominence of numerical diffusion effects when the resisting physical forces are weaker, including cases where the cohesion is low.

**Q25** — Lines 355-358: The statement that "identifying when/whether avalanches are arrested is of paramount importance for, e.g., hazard zoning or designing mitigation measures" is too general. Such a claim should only be made when there is a clear understanding of how these arrest criteria impact key variables, such as runout distance and pressure. Additionally, it is important to note that models are primarily used to generate scenarios, which are then compared to historical data and evaluated by avalanche experts before the final danger assessment.

**A25** — We agree with this suggestion. We propose incorporating your comment when revising this statement, to emphasise that running depth-averaged analyses form only part of the overall process of evaluating risk, and hence potentially designing mitigation measures, and that expert input considering historical data is also required. We shall also note that further work needs to be performed to consider runout distance and hydrostatic/hydrodynamic components of pressure.